# Normalization-Equivariant Neural Networks with Application to Image Denoising

**Sébastien Herbreteau**      **Emmanuel Moebel**

**Charles Kervrann**
Centre Inria de l'Université de Rennes, France
`{sebastien.herbreteau, emmanuel.moebel, charles.kervrann}@inria.fr`

## Abstract

In many information processing systems, it may be desirable to ensure that any change of the input, whether by shifting or scaling, results in a corresponding change in the system response. While deep neural networks are gradually replacing all traditional automatic processing methods, they surprisingly do not guarantee such normalization-equivariance (scale + shift) property, which can be detrimental in many applications. To address this issue, we propose a methodology for adapting existing neural networks so that normalization-equivariance holds by design. Our main claim is that not only ordinary convolutional layers, but also all activation functions, including the ReLU (rectified linear unit), which are applied element-wise to the pre-activated neurons, should be completely removed from neural networks and replaced by better conditioned alternatives. To this end, we introduce affine-constrained convolutions and channel-wise sort pooling layers as surrogates and show that these two architectural modifications do preserve normalization-equivariance without loss of performance. Experimental results in image denoising show that normalization-equivariant neural networks, in addition to their better conditioning, also provide much better generalization across noise levels.

## 1 Introduction

Sometimes wrongly confused with the invariance property which designates the characteristic of a function $f$ not to be affected by a specific transformation $\mathcal{T}$ applied beforehand, the equivariance property, on the other hand, means that $f$ reacts in accordance with $\mathcal{T}$. Formally, invariance is $f \circ \mathcal{T} = f$ whereas equivariance reads $f \circ \mathcal{T} = \mathcal{T} \circ f$, where $\circ$ denotes the function composition operator. Both invariance and equivariance play a crucial role in many areas of study, including physics, computer vision, signal processing and have recently been studied in various settings for deep learning-based models [3, 6, 10, 13, 14, 16, 21, 23, 31, 38, 40, 41, 43].

In this paper, we focus on the equivariance of neural networks $f_\theta$ to a specific transformation $\mathcal{T}$, namely normalization. Although highly desirable in many applications and in spite of its omnipresence in machine learning, current neural network architectures do not equivary to normalization. With application to image denoising, for which *normalization-equivariance* is generally guaranteed for a lot of conventional methods [5, 18, 20, 37], we propose a methodology for adapting existing neural networks, and in particular denoising CNNs [8, 30, 46–48], so that *normalization-equivariance* holds by design. In short, the proposed adaptation is based on two innovations:

1. affine convolutions: the weights from one layer to each neuron from the next layer, *i.e.* the convolution kernels in a CNN, are constrained to encode affine combinations of neurons (the sum of the weights is equal to $1$).

37th Conference on Neural Information Processing Systems (NeurIPS 2023).

2. channel-wise sort pooling: all activation functions that apply element-wise, such as the ReLU, are substituted with higher-dimensional nonlinearities, namely two by two sorting along channels that constitutes a fast and efficient *normalization-equivariant* alternative.

Despite strong architectural constraints, we show that these simple modifications do not degrade performance and, even better, increase robustness to noise levels in image denoising both in practice and in theory.

## 2   Related Work

A non-exhaustive list of application fields where equivariant neural networks were studied includes graph theory, point cloud analysis and image processing. Indeed, graph neural networks are usually expected to equivary, in the sense that a permutation of the nodes of the input graph should permute the output nodes accordingly. Several specific architectures were investigated to guarantee such a property [3, 21, 38]. In parallel, rotation and translation-equivariant networks for dealing with point cloud data were proposed in a recent line of research [6, 13, 40]. A typical application is the ability for these networks to produce direction vectors consistent with the arbitrary orientation of the input point clouds, thus eliminating the need for data augmentation. Finally, in the domain of image processing, it may be desirable that neural networks produce outputs that equivary with regard to rotations of the input image, whether these outputs are vector fields [31], segmentation maps [41, 43], or even bounding boxes for object tracking [16].

In addition to their better conditioning, equivariant neural networks by design are expected to be more robust to outliers. A spectacular example has been revealed by S. Mohan *et al.* [33] in the field of image denoising. By simply removing the additive constant ("bias") terms in neural networks with ReLU activation functions, they showed that a much better generalization at noise levels outside the training range was ensured. Although they do not fully elucidate why biases prevent generalization, and their removal allows it, the authors establish some clues that the answer is probably linked to the *scale-equivariant* property of the resulting encoded function: rescaling the input image by a positive constant value rescales the output by the same amount.

## 3   Overview of normalization-equivariance

### 3.1   Definitions and properties of three types of fundamental equivariances

We start with formal definitions of the different types of equivariances studied in this paper. Please note that our definition of "scale" and "shift" may differ from the definitions given by some authors in the image processing literature.

**Definition 1** *A function $f : \mathbb{R}^n \mapsto \mathbb{R}^m$ is said to be:*

- *scale-equivariant if $\forall x \in \mathbb{R}^n, \forall \lambda \in \mathbb{R}_*^+,\ f(\lambda x) = \lambda f(x)$,*
- *shift-equivariant if $\forall x \in \mathbb{R}^n, \forall \mu \in \mathbb{R},\ f(x + \mu) = f(x) + \mu$,*
- *normalization-equivariant if it is both scale-equivariant and shift-equivariant:*

$$\forall x \in \mathbb{R}^n, \forall \lambda \in \mathbb{R}_*^+, \forall \mu \in \mathbb{R},\ f(\lambda x + \mu) = \lambda f(x) + \mu,$$

*where addition with the scalar shift $\mu$ is applied element-wise.*

Note that the *scale-equivariance* property is more often referred to as positive homogeneity in pure mathematics. Like linear maps that are completely determined by their values on a basis, the above described equivariant functions are actually entirely characterized by the values their take on specific subsets of $\mathbb{R}^n$, as stated by the following lemma (see proof in Appendix C.1).

**Lemma 1 (Characterizations)** *$f : \mathbb{R}^n \mapsto \mathbb{R}^m$ is entirely determined by its values on the:*

- *unit sphere $\mathcal{S}$ of $\mathbb{R}^n$ if it is scale-equivariant,*
- *orthogonal complement of $\mathrm{Span}(\mathbf{1}_n)$, i.e. $\mathrm{Span}(\mathbf{1}_n)^\perp$, if it is shift-equivariant,*
- *intersection $\mathcal{S} \cap \mathrm{Span}(\mathbf{1}_n)^\perp$ if it is normalization-equivariant,*

*where $\mathbf{1}_n$ denotes the all-ones vector of $\mathbb{R}^n$.*

Finally, Lemma 2 highlights three basic equivariance-preserving mathematical operations that can be used as building blocks for designing neural network architectures (see proof in Appendix C.1).

**Lemma 2 (Operations preserving equivariance)** *Let $f$ and $g$ be two equivariant functions of the same type (either in scale, shift or normalization). Then, subject to dimensional compatibility, all of the following functions are still equivariant:*

- $f \circ g$ *($f$ composed with $g$),*
- $x \mapsto (f(x)^\top \; g(x)^\top)^\top$ *(concatenation of $f$ and $g$),*
- $(1-t)f + tg$ *for all $t \in \mathbb{R}$ (affine combination of $f$ and $g$).*

## 3.2 Examples of normalization-equivariant conventional denoisers

A ("blind") denoiser is basically a function $f : \mathbb{R}^n \mapsto \mathbb{R}^n$ which, given a noisy image $y \in \mathbb{R}^n$, tries to map the corresponding noise-free image $x \in \mathbb{R}^n$. Since scaling up an image by a positive factor $\lambda$ or adding it up a constant shift $\mu$ does not change its contents, it is natural to expect *scale* and *shift equivariance*, *i.e. normalization equivariance*, from the denoising procedure emulated by $f$. In image denoising, a majority of methods usually assume an additive white Gaussian noise model with variance $\sigma^2$. The corruption model then reads $y \sim \mathcal{N}(x, \sigma^2 I_n)$, where $I_n$ denotes the identity matrix of size $n$, and the noise standard deviation $\sigma > 0$ is generally passed as an additional argument to the denoiser ("non-blind" denoising). In this case, the augmented function $f : (y, \sigma) \in \mathbb{R}^n \times \mathbb{R}_*^+ \mapsto \mathbb{R}^n$ is said *normalization-equivariant* if:

$$\forall (y, \sigma) \in \mathbb{R}^n \times \mathbb{R}_*^+, \forall \lambda \in \mathbb{R}_*^+, \forall \mu \in \mathbb{R}, \; f(\lambda y + \mu, \lambda\sigma) = \lambda f(y, \sigma) + \mu, \tag{1}$$

as, according to the laws of statistics, $\lambda y + \mu \sim \mathcal{N}(\lambda x + \mu, (\lambda\sigma)^2 I_n)$. In what follows, we give some well-known examples of traditional denoisers that are *normalization-equivariant* (see proofs in Appendix C.2).

**Noise-reduction filters:** The most rudimentary methods for image denoising are the smoothing filters, among which we can mention the averaging filter or the Gaussian filter for the linear filters and the median filter which is nonlinear. These elementary "blind" denoisers all implement a *normalization-equivariant* function. More generally, one can prove that a linear filter is *normalization-equivariant* if and only if its coefficients add up to $1$. In others words, *normalization-equivariant* linear filters process images by affine combinations of pixels.

**Patch-based denoising:** The popular N(on)-L(ocal) M(eans) algorithm [5] and its variants [12, 20, 27] consist in computing, for each pixel, an average of its neighboring noisy pixels, weighted by the degree of similarity of the patches to which they belong. In other words, they process images by convex combinations of pixels. More precisely, NLM can be defined as:

$$f_{\mathrm{NLM}}(y, \sigma)_i = \frac{1}{W_i} \sum_{y_j \in \Omega(y_i)} e^{-\frac{\|p(y_i) - p(y_j)\|_2^2}{h^2}} y_j \quad \text{with} \quad W_i = \sum_{y_j \in \Omega(y_i)} e^{-\frac{\|p(y_i) - p(y_j)\|_2^2}{h^2}} \tag{2}$$

where $y_i$ denotes the $i^{th}$ component of vector $y$, $\Omega(y_i)$ is the set of its neighboring pixels, $p(y_i)$ represents the vectorized patch centered at $y_i$, and the smoothing parameter $h$ is proportional to $\sigma$ as proposed by several authors [4, 12, 29]. Defined as such, $f_{\mathrm{NLM}}$ is a *normalization-equivariant* function. More recently, N(on)-L(ocal) Ridge [18] proposes to process images by linear combinations of similar patches and achieves state-of-the-art performance in unsupervised denoising. When restricting the coefficients of the combinations to sum to $1$, that is imposing affine combination constraints, the resulting algorithm encodes a *normalization-equivariant* function as well.

**TV denoising:** Total variation (TV) denoising [37] is finally one of the most famous image denoising algorithm, appreciated for its edge-preserving properties. In its original form [37], a TV denoiser is defined as a function $f : \mathbb{R}^n \times \mathbb{R}_*^+ \mapsto \mathbb{R}^n$ that solves the following equality-constrained problem:

$$f_{\mathrm{TV}}(y, \sigma) = \arg\min_{x \in \mathbb{R}^n} \|x\|_{\mathrm{TV}} \quad \text{s.t.} \quad \|y - x\|_2^2 = n\sigma^2 \tag{3}$$

where $\|x\|_{\mathrm{TV}} := \|\nabla x\|_2$ is the total variation of $x \in \mathbb{R}^n$. Defined as such, $f_{\mathrm{TV}}$ is a *normalization-equivariant* function.

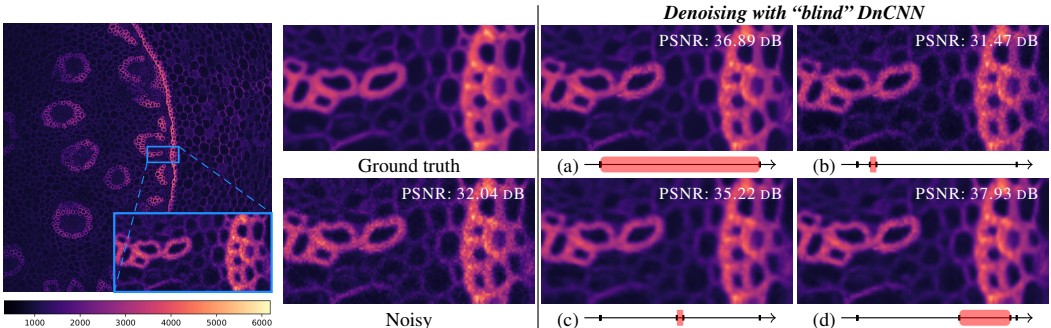

Figure 1: Influence of normalization for deep-learning-based image denoising. The raw input data is a publicly available real-world noisy image of the *Convallaria* dataset [36]. "Blind" DnCNN [47] with official pre-trained weights is used for denoising and is applied on four different normalization intervals displayed in red, each of which being included in $[0, 1]$ over which it was learned. PSNR is calculated with the average of 100 independent noisy static acquisitions of the same sample (called ground truth). Interestingly, the straightforward interval $[0, 1]$ does not give the best results. Normalization intervals are (a) $[0, 1]$, (b) $[0.08, 0.12]$, (c) $[0.48, 0, 52]$ and (d) $[0.64, 0.96]$. In the light of the denoising results $(b)$-$(c)$ and $(b)$-$(d)$, DnCNN is neither *shift-equivariant*, nor *scale-equivariant*.

### 3.3 The case of neural networks

Deep learning hides a subtlety about normalization equivariance that deserves to be highlighted. Usually, the weights of neural networks are learned on a training set containing data all normalized to the same arbitrary interval $[a_0, b_0]$. This training procedure improves the performance and allows for more stable optimization of the model. At inference, unseen data are processed within the interval $[a_0, b_0]$ via a $a$-$b$ linear normalization with $a_0 \leq a < b \leq b_0$ denoted $\mathcal{T}_{a,b}$ and defined by:

$$\mathcal{T}_{a,b} : y \mapsto (b - a)\frac{y - \min(y)}{\max(y) - \min(y)} + a\,. \tag{4}$$

Note that this transform is actually the unique linear one with positive slope that exactly bounds the output to $[a, b]$. The data is then passed to the trained network and its response is finally returned to the original range via the inverse operator $\mathcal{T}_{a,b}^{-1}$. This proven pipeline is actually relevant in light of the following proposition.

**Proposition 1** $\forall a < b \in \mathbb{R}, \forall f : \mathbb{R}^n \mapsto \mathbb{R}^m, \mathcal{T}_{a,b}^{-1} \circ f \circ \mathcal{T}_{a,b}$ *is a normalization-equivariant function.*

While normalization-equivariance appears to be solved, a question is still remaining: how to choose the hyperparameters $a$ and $b$ for a given function $f$ ? Obviously, a natural choice for neural networks is to take the same parameters $a$ and $b$ as in the learning phase whatever the input image is, *i.e.* $a = a_0$ and $b = b_0$, but are they really optimal? The answer to this question is generally negative. Figure 1 depicts an example of the phenomenon in image denoising, taken from a real-world application. In this example, the straightforward choice is largely sub-optimal. This suggests that there are always inherent performance leaks for deep neural networks due to the two degrees of freedom induced by the normalization (*i.e.*, choice of $a$ and choice of $b$). In addition, this poor conditioning can be a source of confusion and misinterpretation in critical applications.

### 3.4 Categorizing image denoisers

Table 1 summarizes the equivariance properties of several popular denoisers, either conventional [5, 11, 15, 18, 37, 44] or deep learning-based [24, 26, 45–47]. Interestingly, if *scale-equivariance* is generally guaranteed for traditional denoisers, not all of them are equivariant to shifts. In particular, the widely used algorithms DCT [44] and BM3D [11] are sensitive to offsets, mainly because the hard thresholding function at their core is not *shift-equivariant*. Regarding the deep-learning-based networks, only DRUNet [46] is insensitive to scale because it is a bias-free convolutional neural

Table 1: Equivariance properties of several image denoisers (left: traditional, right: learning-based)

| | TV | NLM | NLR | DCT | BM3D | WNNM | DnCNN | NLRN | SwinIR | Restormer | DRUNet |
|---|---|---|---|---|---|---|---|---|---|---|---|
| Scale | ✓ | ✓ | ✓ | ✓ | ✓ | ✓ | ✗ | ✗ | ✗ | ✗ | ✓ |
| Shift | ✓ | ✓ | ✓ | ✗ | ✗ | ✗ | ✗ | ✗ | ✗ | ✗ | ✗ |

network with only ReLU activation functions [33]. In particular, all transformer models [7, 24, 26, 45], even bias-free, are not *scale-equivariant* due to their inherent attention-based modules. In the next section, we show how to adapt existing neural architectures to guarantee *normalization-equivariance* without loss of performance and study the resulting class of parameterized functions $(f_\theta)$.

## 4    Design of Normalization-Equivariant Networks

### 4.1    Affine convolutions

To justify the introduction of a new type of convolutional layers, let us study one of the most basic neural network, namely the linear (parameterized) function $f_\Theta : x \in \mathbb{R}^n \mapsto \Theta x$, where parameters $\Theta$ are a matrix of $\mathbb{R}^{m \times n}$. Indeed, $f_\Theta$ can be interpreted as a dense neural network with no bias, no hidden layer and no activation function. Obviously, $f_\Theta$ is always *scale-equivariant*, whatever the weights $\Theta$. As for the *shift-equivariance*, a simple calculation shows that:

$$x \mapsto \Theta x \text{ is } \textit{shift-equivariant} \Leftrightarrow \forall x \in \mathbb{R}^n, \forall \mu \in \mathbb{R}, \Theta(x + \mu \mathbf{1}_n) = \Theta x + \mu \mathbf{1}_m \Leftrightarrow \Theta \mathbf{1}_n = \mathbf{1}_m . \quad (5)$$

Therefore, $f_\Theta$ is *normalization-equivariant* if and only if each row of matrix $\Theta$ sums to 1. In other words, for the *normalization-equivariance* to hold, the rows of $\Theta$ must encode weights of affine combinations. Transposing the demonstration to any convolutional neural network follows from the observation that a convolution from an input layer of size $H \times W \times C$ to an output layer of size $H' \times W' \times C'$ can always be represented with a dense connection by vectorizing the input and output layers. The elements of the $C'$ convolutional kernels of size $k \times k \times C$ each are then stored separately along the rows of the (sparse) transition matrix $\Theta$ of size $(H' \times W' \times C') \times (H \times W \times C)$. Therefore, a convolutional layer preserves the *normalization-equivariance* if and only if the weights of the each convolutional kernel sums to 1. In the following, we call such convolutional layers "affine convolutions".

In order to guarantee the affine constraint on each convolutional kernel throughout the training phase, one possibility is to "telescope" the circular shifted version of an unconstrained kernel to itself (this way, the sum of the resulting trainable coefficients cancels out) and then add the inverse of the kernel size element-wise as a non-trainable offset. Despite this over-parameterized form (involving an extra degree of freedom), we found this solution to be easier to use in practice. Moreover, it ensures that all coefficients of the affine kernels follow the same law at initialization.

Since *normalization-equivariance* is preserved through function composition, concatenation and affine combination (see Lemma 2), a (linear) convolutional neural network composed of only affine convolutions with no bias and possibly skip or *affine* residual connections (trainable affine combination of two layers), is guaranteed to be *normalization-equivariant*, provided that padding is performed with existing features (reflect, replicate or circular padding for example). Obviously, in their current state, these neural networks are of little interest, as linear functions do not encode best-performing functions for many applications, image denoising being no exception. Nevertheless, based on such networks, we show in the next subsection how to introduce nonlinearities without breaking the *normalization-equivariance*.

### 4.2    Channel-wise sort pooling as a normalization-equivariant alternative to ReLU

The first idea that comes to mind is to apply a nonlinear activation function $\varphi : \mathbb{R} \mapsto \mathbb{R}$ preserving *normalization-equivariance* after each affine convolution. In other words, we look for a nonlinear solution $\varphi$ of the characteristic functional equation of *normalization-equivariant* functions (see Def. 1) for $n = 1$. Unfortunately, according to Prop. 2 (see proof in Appendix C.1 which is based on Lemma 1), the unique solution is the identity function which is linear. Therefore, activation functions that apply element-wise are to be excluded.

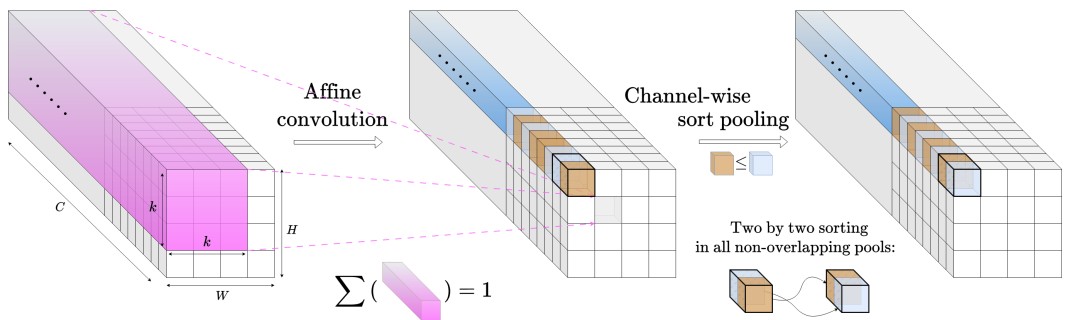

Figure 2: Illustration of the proposed alternative for replacing the traditional scheme "convolution + element-wise activation function" in convolutional neural networks: affine convolutions supersede ordinary ones by restricting the coefficients of each kernel to sum to one and the proposed sort pooling patterns introduce nonlinearities by sorting two by two the pre-activated neurons along the channels.

**Proposition 2** *Let* $\mathrm{NE}(n)$ *be the set of normalization-equivariant functions from* $\mathbb{R}^n$ *to* $\mathbb{R}^n$.
$\mathrm{NE}(1) = \{x \mapsto x\}$ *and*

$$\mathrm{NE}(2) = \left\{ (x_1, x_2) \mapsto A \begin{pmatrix} x_1 \\ x_2 \end{pmatrix} \text{ if } x_1 \leq x_2 \text{ else } B \begin{pmatrix} x_1 \\ x_2 \end{pmatrix} \,\middle|\, A, B \in \mathbb{R}^{2 \times 2} \text{ s.t. } A\mathbf{1}_2 = B\mathbf{1}_2 = \mathbf{1}_2 \right\}.$$

To find interesting nonlinear functions, one needs to examine multi-dimensional activation functions, *i.e.* ones of the form $\varphi : \mathbb{R}^n \mapsto \mathbb{R}^m$ with $n \geq 2$. In order to preserve the dimensions of the neural layers and to limit the computational costs, we focus on the case $n = m = 2$, meaning that $\varphi$ processes pre-activated neurons by pairs. According to Prop. 2, the *normalization-equivariant* functions from $\mathbb{R}^2$ to $\mathbb{R}^2$ are parameterized by two matrices $A, B \in \mathbb{R}^{2 \times 2}$ such that $A\mathbf{1}_2 = B\mathbf{1}_2 = \mathbf{1}_2$ and apply a different (affine-constrained) linear mapping depending on whether or not the input is in ascending order. As long as $A \neq B$, the resulting function is nonlinear and makes it *de facto* a candidate to replace the conventional one-dimensional activation functions such as the popular ReLU (rectified linear unit) function. Interestingly, when arbitrarily choosing $A$ and $B$ to be the permutation matrices of $\mathbb{R}^{2 \times 2}$, the resulting *normalization-equivariant* function simply reads:

$$\varphi : (x_1, x_2) \in \mathbb{R}^2 \mapsto \begin{pmatrix} \min(x_1, x_2) \\ \max(x_1, x_2) \end{pmatrix}, \tag{6}$$

which is nothing else than the sorting function in $\mathbb{R}^2$. Clearly, it is among the simplest *normalization-equivariant* nonlinear function from $\mathbb{R}^2$ to $\mathbb{R}^2$ and it is the one we consider as surrogate for the one-dimensional activation functions (choosing other functions, that is considering other choices for $A$ and $B$, does not bring improvements in terms of performance in our experiments). More generally, it is easy to show that all the sorting functions of $\mathbb{R}^n$ are *normalization-equivariant* and are nonlinear as soon as $n \geq 2$. Note that such sorting operators have been promoted by [2, 9] in totally different contexts for their norm-preserving properties of the backpropagated gradients.

Since the sorting function (6) is to be applied on non-overlapping pairs of neurons, the partitioning of layers needs to be determined. In order not to mix unrelated neurons, we propose to apply this two-dimensional activation function channel-wisely across layers and call this operation "sort pooling" in reference to the max pooling operation, widely used for downsampling, and from which it can be effectively implemented. Figure 2 illustrates the sequence of the two proposed innovations, namely affine convolution followed by channel-wise sort pooling, to replace the traditional scheme "conv+ReLU", while guaranteeing *normalization-equivariance*.

### 4.3 Encoding adaptive affine filters

Based on Lemma 2, we can formulate the following proposition which tells more about the class of parameterized functions $(f_\theta)$ encoded by the proposed networks.

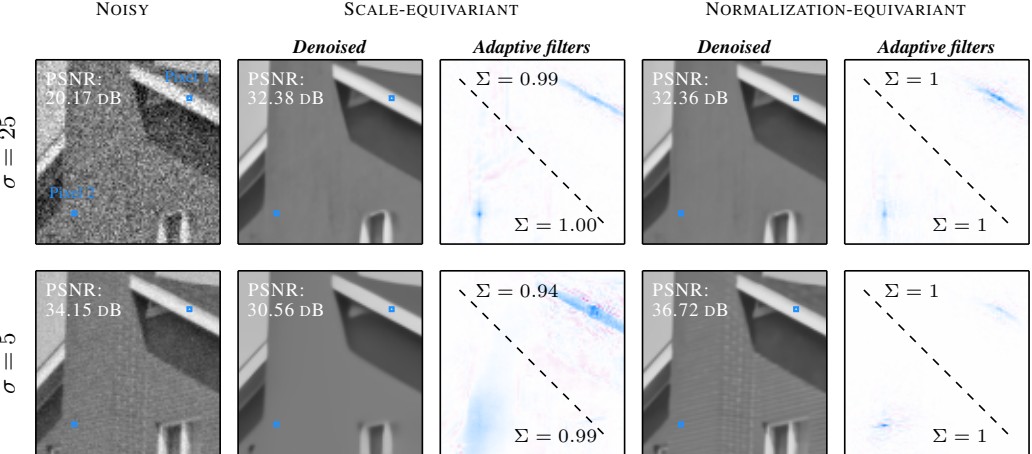

Figure 3: Visual comparisons of the generalization capabilities of a *scale-equivariant* neural network (left) and its *normalization-equivariant* counterpart (right) for Gaussian noise. Both networks were trained for Gaussian noise at noise level $\sigma = 25$ exclusively. The adaptive filters (rows of $A_\theta^{y_r}$ in Prop. 3) are indicated for two particular pixels as well as the sum of their coefficients (note that some weights are negative, indicated in red). The *scale-equivariant* network tends to excessively smooth out the image when evaluated at a lower noise level, whereas the *normalization-equivariant* network is more adaptable and considers the underlying texture to a greater extent.

**Proposition 3** *Let $f_\theta^{NE} : \mathbb{R}^n \mapsto \mathbb{R}^m$ be a CNN composed of only:*

- *affine convolution kernels with no bias and where padding is made of existing features,*
- *sort pooling nonlinearities,*
- *possibly skip or affine residual connections, and max or average pooling layers.*

*Then, $f_\theta^{NE}$ is a normalization-equivariant continuous piecewise-linear function with finitely many pieces. Moreover, on each piece represented by the vector $y_r$,*

$$f_\theta^{NE}(y) = A_\theta^{y_r} y, \quad with \ A_\theta^{y_r} \in \mathbb{R}^{m \times n} \ such \ that \ A_\theta^{y_r} \mathbf{1}_n = \mathbf{1}_m \,.$$

In Prop. 3, the subscripts on $A_\theta^{y_r}$ serve as a reminder that this matrix depends on the sort pooling activation patterns, which in turn depend on both the input vector $y$ and the weights $\theta$. As already revealed for bias-free networks with ReLU [33], $A_\theta^{y_r}$ is the Jacobian matrix of $f_\theta^{NE}$ taken at any point $y$ in the interior of the piece represented by vector $y_r$. Moreover, as $A_\theta^{y_r} \mathbf{1}_n = \mathbf{1}_m$, the output vector of such networks are locally made of fixed affine combinations of the entries of the input vector. And since a CNN has a limited receptive field centered on each pixel, $f_\theta^{NE}$ can be thought of as an adaptive filter that produces an estimate of each pixel through a custom affine combination of pixels. By examining these filters in the case of image denoising (see Fig. 3), it becomes apparent that they vary in their characteristics and are intricately linked to the contents of the underlying images. Indeed, these filters are specifically designed to cater to the specific local features of the noisy image: averaging is done over uniform areas without affecting the sharpness of edges. Note that this behavior has already been extensively studied by [33] for unconstrained filters.

The total number of fixed adaptive affine filters depends on the weights $\theta$ of the network $f_\theta^{NE}$ and is bounded by $2^S$ where $S$ represents the total number of sort pooling patterns traversed to get from the receptive filed to its final pixel (assuming no max pooling layers). Obviously, this upper bound grows exponentially with $S$, suggesting that a limited number of sort pooling operations may generate an extremely large number of filters. Interestingly, if ReLU activation functions where used instead, the upper bound would reach $2^{2S}$.

## 5   Experimental results

We demonstrate the effectiveness and versatility of the proposed methodology in the case of image denoising. To this end, we modify two well-established neural network architectures for image denoising, chosen for both their simplicity and efficiency, namely DRUNet [46]: a state-of-the-art

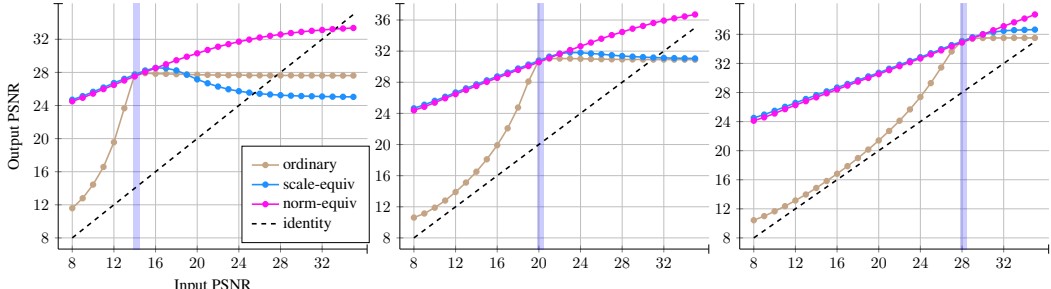

Figure 4: Comparison of the performance of our *normalization-equivariant* alternative with its *scale-equivariant* and *ordinary* counterparts for Gaussian denoising with the same architecture on Set12 dataset. The vertical blue line indicates the unique noise level on which the "blind" networks were trained exclusively (from left to right: $\sigma = 50$, $\sigma = 25$ and $\sigma = 10$). In all cases, *normalization-equivariant* networks generalize much more robustly beyond the training noise level.

U-Net with residual connections [17]; and FDnCNN, the unpublished flexible variant of the popular DnCNN [47]: a simple feedforward CNN that chains "conv+ReLU" layers with no downsampling, no residual connections and no batch normalization during training [19], and with a tunable noise level map as additional input [48]. We show that adapting these networks to become *normalization-equivariant* does not adversely affect performance and, better yet, increases their generalization capabilities. For each scenario, we train three variants of the original Gaussian denoising network for grayscale images: *ordinary* (original network with additive bias), *scale-equivariant* (bias-free variation with ReLU [33]) and our *normalization-equivariant* architecture (see Fig. 2). Details about training and implementations can be found in Appendix A and Appendix B. Unless otherwise noted, all results presented in this paper are obtained with DRUNet [46]; similar outcomes can be achieved with FDnCNN [47] architecture (see Appendix D).

Finally, note that both DRUNet [46] and FDnCNN [47] can be trained as "blind" but also as "non-blind" denoisers and thus achieve increased performance, by passing an additional noisemap as input. In the case of additive white Gaussian noise of variance $\sigma^2$, the noisemap is constant equal to $\sigma \mathbf{1}_n$ and the resulting parameterized functions can then be put mathematically under the form $f_\theta : (y, \sigma) \in \mathbb{R}^n \times \mathbb{R}_*^+ \mapsto \mathbb{R}^n$. In order to integrate this feature to *normalization-equivariant* networks as well, a slight modification of the first affine convolutional layer must be made. Indeed, by adapting the proof (5) to the case (1), we can show that the first convolutional layer must be affine with respect to the input image $y$ only – the coefficients of the kernels acting on the image pixels add up to $1$ – while the other coefficients of the kernels need not be constrained.

## 5.1 The proposed architectural modifications do not degrade performance

The performance, assessed in terms of PSNR values, of our *normalization-equivariant* alternative (see Fig. 2) and of its *scale-equivariant* and *ordinary* counterparts is compared in Table 2 for "non-blind" architectures on two popular datasets [32]. We can notice that the performance gap between two different variants is less than 0.05 dB at most for all noise levels, which is not significant. This result suggests that the class of parameterized functions $(f_\theta)$ currently used in image denoising can drastically be reduced at no cost. Moreover, it shows that it is possible to dispense with activation functions, such as the popular ReLU: nonlinearities can simply be brought by sort pooling patterns. In terms of subjective visual evaluation, we can draw the same conclusion since images produced by two architectural variants inside the training range are hardly distinguishable (see Fig. 3 at $\sigma = 25$).

## 5.2 Increased robustness across noise levels

S. Mohan *et al.* [33] revealed that bias-free neural networks with ReLU, which are *scale-equivariant*, could much better generalize when evaluated at new noise levels beyond their training range, than their counterparts with bias that systematically overfit. Even if they do not fully elucidate how such networks achieve this remarkable generalization, they suggest that *scale-equivariance* certainly plays a major role. What about *normalization-equivariance* then? We have compared the robustness faculties of the three variants of networks when trained at a fixed noise level $\sigma$ for Gaussian noise. Figure

Table 2: The PSNR (dB) results of "non-blind" deep learning-based methods applied to popular grayscale datasets corrupted by synthetic white Gaussian noise with $\sigma = 15$, 25 and 50.

| Dataset | | Set12 | BSD68 |
|---|---|---|---|
| Noise level $\sigma$ | | 15 / 25 / 50 | 15 / 25 / 50 |
| | *ordinary* | 33.23 / 30.92 / 27.87 | 31.89 / 29.44 / 26.54 |
| DRUNet [46] | *scale-equiv* | 33.25 / 30.94 / 27.90 | 31.91 / 29.48 / 26.59 |
| | ***norm-equiv*** | 33.20 / 30.90 / 27.85 | 31.88 / 29.45 / 26.55 |
| | *ordinary* | 32.87 / 30.49 / 27.28 | 31.69 / 29.22 / 26.27 |
| FDnCNN [47] | *scale-equiv* | 32.85 / 30.49 / 27.29 | 31.67 / 29.20 / 26.25 |
| | ***norm-equiv*** | 32.85 / 30.50 / 27.27 | 31.69 / 29.22 / 26.25 |

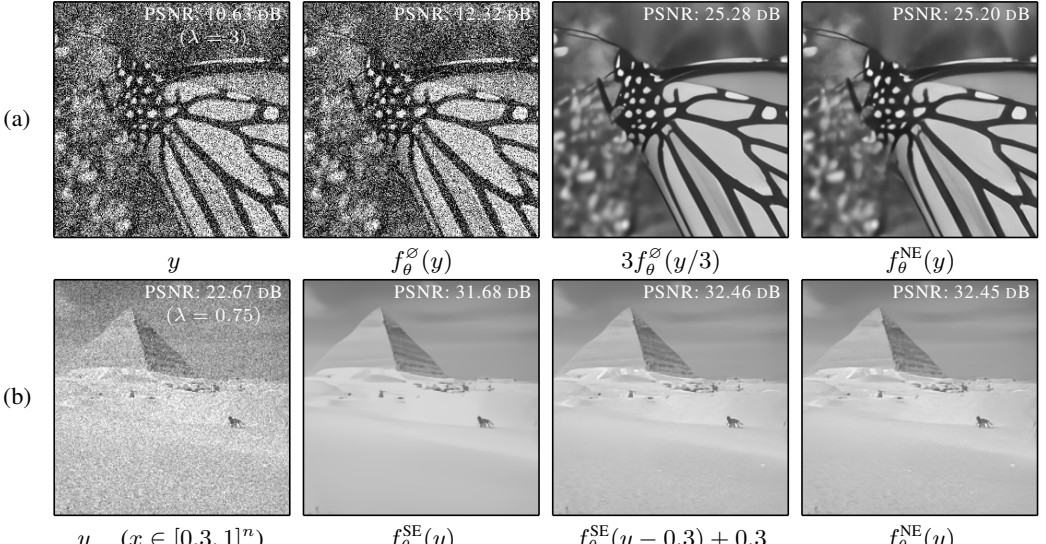

Figure 5: Denoising results for example images of the form $y = x + \lambda\varepsilon$ (see notations of subsection 5.2) with $\sigma = 25/255$ and $x \in [0,1]^n$ by "blind" CNNs specialized for noise level $\sigma$ only. $f_\theta^\varnothing$, $f_\theta^{\text{SE}}$ and $f_\theta^{\text{NE}}$ denote the *ordinary*, *scale-equivariant* and *normalization-equivariant* variants, respectively. In order to get the best results with $f_\theta^\varnothing$ and $f_\theta^{\text{SE}}$, it is necessary know the renormalization parameters $(\lambda, \mu)$ such that $(x - \mu)/\lambda$ belongs to $\mathcal{D} \subset [0,1]^n$ (see subsection 5.2). Note that for $f_\theta^{\text{SE}}$, it is however sufficient to know only $\mu$ as $\lambda$ is implicit by construction. In contrast, $f_\theta^{\text{NE}}$ can be applied directly.

4 summarizes the explicit results obtained: *normalization-equivariance* pushes generalization capabilities of neural networks one step further. While performance is identical to their *scale-equivariant* counterparts when evaluated at higher noise levels, the *normalization-equivariant* networks are, however, much more robust at lower noise levels. This phenomenon is also illustrated in Fig. 3.

**Demystifying robustness** Let $x$ be a clean patch of size $n$, representative of the training set on which a CNN $f_\theta$ was optimized to denoise its noisy realizations $y = x + \varepsilon$ with $\varepsilon \sim \mathcal{N}(0, \sigma^2 I_n)$ (denoising at a fixed noise level $\sigma$ exclusively). Formally, we note $x \in \mathcal{D} \subset \mathbb{R}^n$, where $\mathcal{D}$ is the space of representative clean patches of size $n$ on which $f_\theta$ was trained. We are interested in the output of $f_\theta$ when it is evaluated at $x + \lambda\varepsilon$ (denoising at noise level $\lambda\sigma$) with $\lambda > 0$. Assuming that $f_\theta$ encodes a *normalization-equivariant* function, we have:

$$\forall \lambda \in \mathbb{R}_*^+, \forall \mu \in \mathbb{R}, \ f_\theta(x + \lambda\varepsilon) = \lambda f_\theta((x - \mu)/\lambda + \varepsilon) + \mu \,. \tag{7}$$

The above equality shows how such networks can deal with noise levels $\lambda\sigma$ different from $\sigma$: *normalization-equivariance* simply brings the problem back to the denoising of an implicitly renormalized image patch with fixed noise level $\sigma$. Note that this artificial change of noise level does not make this problem any easier to solve as the signal-to-noise ratio is preserved by normalization.

Obviously, the denoising result of $x + \lambda\varepsilon$ will be all the more accurate as $(x - \mu)/\lambda$ is a representative patch of the training set. In other words, if $(x - \mu)/\lambda$ can still be considered to be in $\mathcal{D}$, then $f_\theta$ should output a consistent denoised image patch. For a majority of methods [46–48], training is performed within the interval $[0, 1]$ and therefore $x/\lambda$ still belongs generally to $\mathcal{D}$ for $1 < \lambda < 10$ (contraction), but this is much less true for $\lambda < 1$ (stretching) for the reason that it may exceed the bounds of the interval $[0, 1]$. This explains why *scale-equivariant* functions do not generalize well to noise levels lower than their training one. In contrast, *normalization-equivariant* functions can benefit from the implicit extra adjustment parameter $\mu$. Indeed, there exists some cases where the stretched patch $x/\lambda$ is not in $\mathcal{D}$ but $(x - \mu)/\lambda$ is (see Fig. 5b). This is why *normalization-equivariant* networks are more able to generalize at low noise levels. Note that, based on this argument, *ordinary* neural networks trained at a fixed noise level $\sigma$ can also be used to denoise images at noise level $\lambda\sigma$, provided that a correct normalization is done beforehand [42]. However, this time the normalization is explicit: the exact scale factor $\lambda$, and possibly the shift $\mu$, must be known (see Fig. 5a).

It turns out that this theoretical argument is valid for a wide range of noise types, not only Gaussian noise. Indeed, the same argument holds for any additive noise $\varepsilon$ that possesses the scaling property: $\lambda\varepsilon$ belongs to the same family of probability distributions as $\varepsilon$ (e.g., Gaussian, uniform, Laplace or even Rayleigh noise which is not zero-mean). By the way, the authors of [33] had already verified the noise generalization capabilities of *scale-equivariant* networks for uniform noise in addition to Gaussian noise, without fully elucidating why it works. In Appendix D, we checked experimentally that "blind" *normalization-equivariant* networks trained on additive uniform, Laplace or Rayleigh noise at a single noise level are much more robust at unseen noise levels than their *scale-equivariant* and *ordinary* counterparts.

## 6    Conclusion and perspectives

In this work, we presented an original approach to adapt the architecture of existing neural networks so that they become *normalization-equivariant*, a property highly desirable and expected in many applications such that image denoising. We argue that the classical pattern "conv+ReLU" can be favorably replaced by the two proposed innovations: affine convolutions that ensure that all coefficients of the convolutional kernels sum to one; and channel-wise sort pooling nonlinearities as a substitute for all activation functions that apply element-wise, including ReLU or sigmoid functions. Despite these two important architectural changes, we show that the performance of these alternative networks is not affected in any way. On the contrary, thanks to their better-conditioning, they benefit, in the context of image denoising, from an increased interpretability and especially robustness to variable noise levels both in practice and in theory.

## Limitations

We would like to mention that the proposed architectural modifications for enforcing *normalization-equivariance* require a longer training for achieving comparable performance with its original counterparts (see Appendix B), and may be incompatible with some specific network layers such as batch-norm [19] or attention-based modules [7, 24, 26, 45]. Moreover, our method has shown its potential mainly to image denoising as it stands, even though in principle *normalization-equivariance* may be applicable and helpful in other tasks as well (see preliminary results about image classification in Appendix D). Discovering similar advantages of *normalization-equivariance* in other computer vision tasks, possibly related to outlier robustness, is an interesting avenue of research for future work.

## Acknowledgments and Disclosure of Funding

This work was supported by Bpifrance agency (funding) through the LiChIE contract. Computations were performed on the Inria Rennes computing grid facilities partly funded by France-BioImaging infrastructure (French National Research Agency - ANR-10-INBS-04-07, "Investments for the future"). We would like to thank R. Fraisse (Airbus) for fruitful discussions.

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

# A Description of the denoising architectures and implementation

## A.1 Description of models

**DRUNet:**  DRUNet [46] is a U-Net architecture, and as such has an encoder-decoder type pathway, with residual connections [17]. Spatial downsampling is performed using $2 \times 2$ convolutions with stride 2, while spatial upsampling leverages $2 \times 2$ transposed convolutions with stride 2 (which is equivalent to a $1 \times 1$ sub-pixel convolution [39]). The number of channels in each layer from the first scale to the fourth scale are $64$, $128$, $256$ and $512$, respectively. Each scale is composed of 4 successive residual blocks "$3 \times 3$ conv + ReLU + $3 \times 3$ conv".

**FDnCNN:**  FDnCNN [47] is the unpublished flexible variant of the popular DnCNN [47]. It consists of 20 successive $3 \times 3$ convolutional layers with $64$ channels each and ReLU nonlinearities. As opposed to DnCNN, FDnCNN does not use neither batch normalization [19] for training, nor residual connections [17] and can handle an optional noisemap (concatenated with the input noisy image). Note that this architecture does not use downsampling or upsampling. Finally, the authors [47] recommend to train it by minimizing the $\ell_1$ loss instead of the mean squared error (MSE).

## A.2 Description of variants

**Ordinary:**  The *ordinary* variant is built by appending additive constant ("bias") terms after each convolution of the original architecture. Note that the original FDnCNN [47] model is already in the *ordinary* mode.

**Scale-equivariant:**  Since both models (DRUNet and FDnCNN) use only ReLU activation functions, removing all additive constant ("bias") terms is sufficient to ensure *scale-equivariance* [33]. Note that the original DRUNet [46] model is already in the *scale-equivariant* mode.

**Normalization-equivariant:**  All convolutions are replaced by the proposed affine-constrained convolutions without "bias" and with reflect padding, and the proposed channel-wise sort pooling patterns supersede ReLU nonlinearities. Moreover, classical residual connections are replaced by *affine* residual connections (the sum of two layers $l_1$ and $l_2$ is replaced by their affine combination $(1-t)l_1 + tl_2$ where $t$ is a trainable scalar parameter).

## A.3 Practical implementation of normalization-equivariant networks

The channel-wise sort pooling operations can be efficiently implemented by concatenating the sublayer obtained with channel-wise one-dimensional max pooling with kernel size 2 and its counterpart obtained with min pooling. Note that intertwining these two sub-layers to comply with the original definition is not necessary in practice (although performed anyway in our implementation), since the order of the channels in a CNN is arbitrary.

Regarding the implementation of affine convolutions for training, each unconstrained kernel can be in practice "telescoped" with its circular shifted version (this way, the sum of the resulting trainable coefficients cancels out) and then the inverse of the kernel size is added element-wise as a nontrainable offset. Despite this over-parameterized form (involving an extra degree of freedom), we found this solution to be more easy to use in practice. Moreover, it ensures that all coefficients of the affine kernels follow the same law at initialization. Another possibility is to set an arbitrary coefficient of the kernel (the last one for instance) equal to one minus the sum of all the other coefficients. Note that the solution consisting in dividing each kernel coefficient by the sum of all the other coefficients does not work because it generates numerical instabilities as the divisor may be zero, or close to zero.

All our implementations are written in Python and are based on the PyTorch library [35]. The code and pre-trained models can be downloaded here: `https://github.com/sherbret/normalization_equivariant_nn/`.

Table 3: Training parameters. * indicates that it is divided by half every $100,000$ iterations.

| Model | | Batch size | Patch size | Loss function | Learning rate | Number of iterations |
|---|---|---|---|---|---|---|
| DRUNet [46] | *ordinary* | 16 | $128 \times 128$ | $\ell_1$ | 1e-4* | $800,000$ |
| | *scale-equiv* | 16 | $128 \times 128$ | $\ell_1$ | 1e-4* | $800,000$ |
| | *norm-equiv* | 16 | $128 \times 128$ | MSE | 1e-4 | $1,800,000$ |
| FDnCNN [47] | *ordinary* | 128 | $70 \times 70$ | $\ell_1$ | 1e-4 | $500,000$ |
| | *scale-equiv* | 128 | $70 \times 70$ | $\ell_1$ | 1e-4 | $500,000$ |
| | *norm-equiv* | 128 | $70 \times 70$ | MSE | 1e-4 | $900,000$ |

Table 4: Execution time (in seconds) comparison on a training batch of size $16 \times 1 \times 128 \times 128$ for different variants of the same DRUNet architecture [46] (GPU: Quadro RTX 6000, CPU: 2,3 GHz Intel Core i7). The difference in speed between variants is more pronounced on GPU than on CPU.

| | | Affine | SortPool | Backward pass ↓ | Inference pass ↓ |
|---|---|---|---|---|---|
| **GPU** | *(scale-equiv)* | ✗ | ✗ | 0.229 | 0.067 |
| | | ✗ | ✓ | 0.268 | 0.102 |
| | | ✓ | ✗ | 0.344 | 0.083 |
| | *(norm-equiv)* | ✓ | ✓ | 0.386 | 0.122 |

| | Variant | Backward pass ↓ | Inference pass ↓ |
|---|---|---|---|
| **CPU** | *ordinary* | 38.4 | 12.5 |
| | *scale-equiv* | 37.4 | 12.2 |
| | *norm-equiv* | 45.3 | 14.3 |

Affine: affine-constrained convolutions with reflect padding and affine residual connections.
SortPool: channel-wise sort pooling nonlinearities instead of ReLU.

# B  Description of datasets and training details

We use the same large training set as in [46] for all the models and all the experiments, composed of $8,694$ images, including $400$ images from the Berkeley Segmentation Dataset BSD400 [32], $4,744$ images from the Waterloo Exploration Database [28], $900$ images from the DIV2K dataset [1], and $2,750$ images from the Flickr2K dataset [25]. This training set is augmented via random vertical and horizontal flips and random $90°$ rotations. The dataset BSD32 [32], composed of the 32 images, is used as validation set to control training and select the best model at the end. Finally, the two datasets Set12 and BSD68 [32], strictly disjoint from the training and validation sets, are used for testing.

All the models $f_\theta$ are optimized by minimizing the average reconstruction error between the denoised images $\hat{x} = f_\theta(x + \varepsilon)$, where $\varepsilon \sim \mathcal{N}(0, \sigma^2 I_n)$, and ground-truths $x$ with Adam algorithm [22]. For "non-blind" models, the noise level $\sigma$ is randomly chosen from $[1, 50]$ during training. The training parameters, specific to each model and its variants, are guided by the instructions of the original papers [46, 47], to the extent possible, and are summarized in Table 3. Note that each training iteration consists in a gradient pass on a batch composed of patches randomly cropped from training images. *Normalization-equivariant* variants need a longer training and always use a constant learning rate (speed improvements are however certainly possible by adapting the learning rate throughout optimization, but we did not investigated much about it). Furthermore, contrary to [46] where the $\ell_1$ loss function is recommended to achieve better performance, supposedly due to its outlier robustness properties, we obtained slightly better results with the usual mean squared error (MSE) loss when dealing with *normalization-equivariant* networks. Training was performed with a Quadro RTX 6000 GPU.

In Table B, we compare the computational costs of different variants for training and inference. Interestingly, the computational cost on GPU for training is much more sensitive to the "affine mode" (involving affine-constrained convolutions with reflect padding and affine residual connection) than to sort pooling nonlinearities, while it is the opposite for inference. All in all, for gaining *normalization-equivariance*, the learning and inference time is almost doubled for the DRUNet architecture [46] on GPU. Note however that we do not claim to have the most optimized implementation and there

is probably room for improvement. Surprisingly, the difference in speed between all variants is much less pronounced on CPU. In particular, the inference pass takes only about $20\%$ longer for the *normalization-equivariant* DRUNet on CPU.

## C  Mathematical proofs for normalization-equivariant neural networks

### C.1  Proofs of Lemmas and Propositions

**Lemma 1**(Characterizations)

**Proof:** For each type of equivariance, both existence and uniqueness of $f$ must be proven. Let $\mathbf{0}_n$ be the zero vector of $\mathbb{R}^n$ and $(y_x)_{x \in \mathcal{C}}$ the values that $f$ takes on its characteristic set $\mathcal{C}$.

*Scale-equivariance:*

- Uniqueness: Let $f$ and $g$ two *scale-equivariant* functions such that $\forall x \in \mathcal{S}, f(x) = g(x)$. First of all, for any *scale-equivariant* function $h$, $h(\mathbf{0}_n) = h(2 \cdot \mathbf{0}_n) = 2h(\mathbf{0}_n)$, hence $h(\mathbf{0}_n) = \mathbf{0}_m$. Therefore, $f(\mathbf{0}_n) = g(\mathbf{0}_n) = \mathbf{0}_m$.

  Let $x \in \mathbb{R}^n \setminus \{\mathbf{0}_n\}$. As $\frac{x}{\|x\|} \in \mathcal{S}$, we have $f(\frac{x}{\|x\|}) = g(\frac{x}{\|x\|}) \Rightarrow \frac{1}{\|x\|}f(x) = \frac{1}{\|x\|}g(x) \Rightarrow f(x) = g(x)$. Finally, $f = g$.

- Existence: Let $f : x \in \mathbb{R}^n \mapsto \begin{cases} \|x\| \cdot y_{\frac{x}{\|x\|}} & \text{if } x \neq \mathbf{0}_n \\ \mathbf{0}_m & \text{otherwise} \end{cases}$. Note that $\forall x \in \mathcal{S}, f(x) = y_x$.

  Let $x \in \mathbb{R}^n$ and $\lambda \in \mathbb{R}_*^+$. If $x \neq \mathbf{0}_n$, $f(\lambda x) = \|\lambda x\| \cdot y_{\frac{\lambda x}{\|\lambda x\|}} = \lambda \|x\| \cdot y_{\frac{x}{\|x\|}} = \lambda f(x)$ and if $x = \mathbf{0}_n$ $f(\lambda x) = \mathbf{0}_m = \lambda f(x)$, hence $f$ is *scale-equivariant*.

*Shift-equivariance:*

- Uniqueness: Let $f$ and $g$ two *shift-equivariant* functions such that $\forall x \in \mathrm{Span}(\mathbf{1}_n)^\perp, f(x) = g(x)$. Let $x \in \mathbb{R}^n$. By orthogonal decomposition of $\mathbb{R}^n$ into $\mathrm{Span}(\mathbf{1}_n)^\perp$ and $\mathrm{Span}(\mathbf{1}_n)$:

$$\exists! \, (x_1, x_2) \in \mathrm{Span}(\mathbf{1}_n)^\perp \times \mathrm{Span}(\mathbf{1}_n), \; x = x_1 + x_2 \,.$$

  Then, $f(x) = f(x_1 + x_2) = f(x_1) + x_2 = g(x_1) + x_2 = g(x_1 + x_2) = g(x)$.

- Existence: Let $f : x \in \mathbb{R}^n \mapsto y_{x_1} + x_2$, where $x = x_1 + x_2$ is the unique decomposition such that $x_1 \in \mathrm{Span}(\mathbf{1}_n)^\perp$ and $x_2 \in \mathrm{Span}(\mathbf{1}_n)$. Note that $\forall x \in \mathrm{Span}(\mathbf{1}_n)^\perp, f(x) = y_x$. Let $x \in \mathbb{R}^n$ and $\mu \in \mathbb{R}$. $f(x + \mu) = y_{x_1} + x_2 + \mu\mathbf{1}_m = f(x) + \mu$ as if $x$ orthogonally decomposes into $x_1 + x_2$ with $x_1 \in \mathrm{Span}(\mathbf{1}_n)^\perp$ and $x_2 \in \mathrm{Span}(\mathbf{1}_n)$, then $x + \mu$ orthogonally decomposes into $x_1 + (x_2 + \mu\mathbf{1}_m)$. $f$ is then *shift-equivariant*.

*Normalization-equivariance:*

- Uniqueness: Let $f$ and $g$ two *normalization-equivariant* functions such that $\forall x \in \mathcal{S} \cap \mathrm{Span}(\mathbf{1}_n)^\perp, f(x) = g(x)$. First, as $f$ and $g$ are *a fortiori scale-equivariant*, $f(\mathbf{0}_n) = g(\mathbf{0}_n) = \mathbf{0}_m$. Let $x \in \mathbb{R}^n \setminus \{\mathbf{0}_n\}$. By orthogonal decomposition of $\mathbb{R}^n$ into $\mathrm{Span}(\mathbf{1}_n)^\perp$ and $\mathrm{Span}(\mathbf{1}_n)$:

$$\exists! \, (x_1, x_2) \in \mathrm{Span}(\mathbf{1}_n)^\perp \times \mathrm{Span}(\mathbf{1}_n), \; x = x_1 + x_2 \,.$$

  If $x_1 = \mathbf{0}_n$, $f(x) = f(\mathbf{0}_n + x_2) = f(\mathbf{0}_n) + x_2 = \mathbf{0}_m + x_2 = x_2$. Likewise, $g(x) = x_2$, hence $f(x) = g(x)$. Else, if $x_1 \neq \mathbf{0}_n$, $f(x) = f(x_1 + x_2) = f(x_1) + x_2 = \|x_1\|f(\frac{x_1}{\|x_1\|}) + x_2 = \|x_1\|g(\frac{x_1}{\|x_1\|}) + x_2 = g(x_1) + x_2 = g(x_1 + x_2) = g(x)$, as $\frac{x_1}{\|x_1\|} \in \mathcal{S} \cap \mathrm{Span}(\mathbf{1}_n)^\perp$. Finally, $f = g$.

- Existence: Let $f : x \in \mathbb{R}^n \mapsto \begin{cases} \|x_1\| \cdot y_{\frac{x_1}{\|x_1\|}} + x_2 & \text{if } x_1 \neq \mathbf{0}_n \\ x_2 & \text{otherwise} \end{cases}$, where $x = x_1 + x_2$ is the unique decomposition such that $x_1 \in \mathrm{Span}(\mathbf{1}_n)^\perp$ and $x_2 \in \mathrm{Span}(\mathbf{1}_n)$. Note that $\forall x \in \mathcal{S} \cap \mathrm{Span}(\mathbf{1}_n)^\perp, f(x) = y_x$. Let $x \in \mathbb{R}^n$, $\lambda \in \mathbb{R}_*^+$ and $\mu \in \mathbb{R}$. $x$ decomposes

orthogonally into $x_1 + x_2$ with $x_1 \in \mathrm{Span}(\mathbf{1}_n)^\perp$ and $x_2 \in \mathrm{Span}(\mathbf{1}_n)$, and we have $f(\lambda x + \mu) = f(\lambda x_1 + (\lambda x_2 + \mu))$, where $\lambda x_1 + (\lambda x_2 + \mu)$ is the orthogonal decomposition of $\lambda x + \mu$ into $\mathrm{Span}(\mathbf{1}_n)^\perp$ and $\mathrm{Span}(\mathbf{1}_n)$.

If $x_1 = \mathbf{0}_n$, then $\lambda x_1 = \mathbf{0}_n$ and $f(\lambda x + \mu) = \lambda x_2 + \mu = \lambda f(x) + \mu$.

Else, if $x_1 \neq \mathbf{0}_n$, then $\lambda x_1 \neq \mathbf{0}_n$, and $f(\lambda x + \mu) = \|\lambda x_1\| \cdot y_{\frac{\lambda x_1}{\|\lambda x_1\|}} + (\lambda x_2 + \mu) = \lambda(\|x_1\| \cdot y_{\frac{x_1}{\|x_1\|}} + x_2) + \mu = \lambda f(x) + \mu$. Finally, $f$ is *normalization-equivariant*. $\qquad \square$

**Lemma 2** (Operations preserving equivariance)

**Proof:** Let $x \in \mathbb{R}^n$, $\lambda \in \mathbb{R}_*^+$ and $\mu \in \mathbb{R}$.

- If $f$ and $g$ are both *scale-equivariant*, $(f \circ g)(\lambda x) = f(g(\lambda x)) = f(\lambda g(x)) = \lambda f(g(x)) = \lambda(f \circ g)(x)$ and if they are both *shift-equivariant*, $(f \circ g)(x + \mu) = f(g(x + \mu)) = f(g(x) + \mu) = f(g(x)) + \mu = (f \circ g)(x) + \mu$.
- Let $h : x \mapsto (f(x)^\top \ g(x)^\top)^\top$. If $f$ and $g$ are both *scale-equivariant*, $h(\lambda x) = (f(\lambda x)^\top \ g(\lambda x)^\top)^\top = (\lambda f(x)^\top \ \lambda g(x)^\top)^\top = \lambda h(x)$ and if they are both *shift-equivariant*, $h(x + \mu) = (f(x + \mu)^\top \ g(x + \mu)^\top)^\top = (f(x)^\top + \mu \ \ g(x)^\top + \mu)^\top = h(x) + \mu$.
- Let $t \in \mathbb{R}$ and $h : x \mapsto (1 - t)f + tg$. If $f$ and $g$ are both *scale-equivariant*, $h(\lambda x) = (1 - t)f(\lambda x) + tg(\lambda x) = (1 - t)\lambda f(x) + t\lambda g(x) = \lambda((1 - t)f(x) + tg(x)) = \lambda h(x)$ and if they are both *shift-equivariant*, $h(x + \mu) = (1 - t)f(x + \mu) + tg(x + \mu) = (1 - t)(f(x) + \mu) + t(g(x) + \mu) = (1 - t)f(x) + tg(x) + (1 - t)\mu + t\mu = h(x) + \mu$. $\square$

**Proposition 1**

**Proof:** Let $a < b \in \mathbb{R}$, $f : \mathbb{R}^n \mapsto \mathbb{R}^m$, $x \in \mathbb{R}^n$, $\lambda \in \mathbb{R}_*^+$ and $\mu \in \mathbb{R}$.

We have $\mathcal{T}_{a,b}(\lambda x + \mu) = (b - a)\frac{\lambda x + \mu - \min(\lambda x + \mu)}{\max(\lambda x + \mu) - \min(\lambda x + \mu)} + a = (b - a)\frac{x - \min(x)}{\max(x) - \min(x)} + a = \mathcal{T}_{a,b}(x)$ (*i.e.* $\mathcal{T}_{a,b}$ is *normalization-invariant*). $\mathcal{T}_{a,b}^{-1}$ denotes the inverse transformation intricately linked to the input $x$ of $\mathcal{T}_{a,b}$ (note that this is an improper notation as $\mathcal{T}_{a,b}$ is not bijective). Thus, if $x$ is the input of $\mathcal{T}_{a,b}$, then $\mathcal{T}_{a,b}^{-1} : y \mapsto (\max(x) - \min(x))\frac{y - a}{b - a} + \min(x)$.

$$
\begin{aligned}
(\mathcal{T}_{a,b}^{-1} \circ f \circ \mathcal{T}_{a,b})(\lambda x + \mu) &= \frac{\max(\lambda x + \mu) - \min(\lambda x + \mu)}{b - a}\left((f \circ \mathcal{T}_{a,b})(\lambda x + \mu) - a\right) + \min(\lambda x + \mu)\,, \\
&= \lambda\frac{\max(x) - \min(x)}{b - a}\left((f \circ \mathcal{T}_{a,b})(\lambda x + \mu) - a\right) + \lambda\min(x) + \mu\,, \\
&= \lambda\left(\frac{\max(x) - \min(x)}{b - a}\left((f \circ \mathcal{T}_{a,b})(x) - a\right) + \min(x)\right) + \mu\,, \\
&= \lambda(\mathcal{T}_{a,b}^{-1} \circ f \circ \mathcal{T}_{a,b})(x) + \mu\,.
\end{aligned}
$$

Finally, $\mathcal{T}_{a,b}^{-1} \circ f \circ \mathcal{T}_{a,b}$ is *normalization-equivariant*. $\qquad \square$

**Proposition 2**

**Proof:** Let $\mathrm{id} : x \in \mathbb{R} \mapsto x$ be the identity function. $\mathrm{id}$ is a *normalization-equivariant* function so $\{x \mapsto x\} \subseteq \mathrm{NE}(1)$. Reciprocally, let $f \in \mathrm{NE}(1)$. By *scale-equivariance*, $f(0) = f(2 \times 0) = 2f(0)$, hence $f(0) = 0$. By *shift-equivariance*, $\forall x \in \mathbb{R}$, $f(x) = f(x + 0) = f(0) + x = x$, hence, $f = \mathrm{id}$. Finally, $\mathrm{NE}(1) \subseteq \{x \mapsto x\}$, hence $\mathrm{NE}(1) = \{x \mapsto x\}$. Note that it is coherent with Lemma 1 which states that $f$ is entirely determined by its values on $\mathcal{S} \cap \mathrm{Span}(\mathbf{1}_n)^\perp$, which reduces to the empty set for $n = 1$.

Let $F = \left\{ (x_1, x_2) \mapsto A\begin{pmatrix} x_1 \\ x_2 \end{pmatrix} \text{ if } x_1 \leq x_2 \text{ else } B\begin{pmatrix} x_1 \\ x_2 \end{pmatrix} \,\middle|\, A, B \in \mathbb{R}^{2 \times 2} \text{ s.t. } A\mathbf{1}_2 = B\mathbf{1}_2 = \mathbf{1}_2 \right\}$ and $f \in F$. Let $x \in \mathbb{R}^2$, $\lambda > 0$ and $\mu \in \mathbb{R}$.

$$
f(\lambda x + \mu) = \begin{cases} A(\lambda x + \mu) & \text{if } \lambda x_1 + \mu \leq \lambda x_2 + \mu \\ B(\lambda x + \mu) & \text{otherwise} \end{cases} = \begin{cases} \lambda Ax + \mu & \text{if } x_1 \leq x_2 \\ \lambda Bx + \mu & \text{otherwise} \end{cases} = \lambda f(x) + \mu,
$$

hence $f \in \mathrm{NE}(2)$.

Reciprocally, let $f \in \text{NE}(2)$. For $n = 2$ and when considering the Euclidean distance, $\mathcal{S} \cap \text{Span}(\mathbf{1}_n)^{\perp} = \{-u, u\}$ with $u = (-1/\sqrt{2}, 1/\sqrt{2})$. Let

$$A = \frac{1}{u_2 - u_1} \begin{pmatrix} u_2 - f(u)_1 & f(u)_1 - u_1 \\ u_2 - f(u)_2 & f(u)_2 - u_1 \end{pmatrix} \text{ and } B = \frac{1}{u_2 - u_1} \begin{pmatrix} u_2 + f(-u)_1 & -f(-u)_1 - u_1 \\ u_2 + f(-u)_2 & -f(-u)_2 - u_1 \end{pmatrix}.$$

Let $g : (x_1, x_2) \mapsto A \begin{pmatrix} x_1 \\ x_2 \end{pmatrix}$ if $x_1 \le x_2$ else $B \begin{pmatrix} x_1 \\ x_2 \end{pmatrix}$. We have $g \in F$ since $A\mathbf{1}_2 = B\mathbf{1}_2 = \mathbf{1}_2$ (in particular $g$ is then *normalization-equivariant*) and $\forall x \in \{-u, u\}, g(x) = f(x)$. According to Lemma 1, $g = f$, hence $f \in F$. Finally, $\text{NE}(2) \subseteq F$, hence $\text{NE}(2) = F$. $\qquad \square$

**Proposition 3**

**Proof:** $f_\theta^{\text{NE}}$ is composed of three types of building blocks of the following form:

- affine convolutions: $a_\Theta : x \in \mathbb{R}^n \mapsto \Theta x$ with $\Theta \in \mathbb{R}^{m \times n}$ subject to $\Theta \mathbf{1}_n = \mathbf{1}_m$,

- sort pooling nonlinearities: $\text{sortpool} : \mathbb{R}^n \mapsto \mathbb{R}^n$,

- max pooling layers: $\text{maxpool} : \mathbb{R}^n \mapsto \mathbb{R}^m$ with $m < n$,

which are assembled using:

- function compositions: $\text{comp}(f, g) \mapsto f \circ g$,

- skip connections: $\text{skip}(f, g) \mapsto (x \mapsto (f(x)^\top \ g(x)^\top)^\top)$,

- affine residual connections: $\text{ares}_t(f, g) \mapsto (1 - t)f + tg$ with $t \in \mathbb{R}$.

Note that the rows of $\Theta$ in $a_\Theta$ encode the convolution kernels in a CNN and the trainable parameters, denoted by $\theta$, are only composed of matrices $\Theta$ and scalars $t$. Moreover, note that average pooling layers are nothing else than affine convolutions with fixed parameters.

Since $a_\Theta$, $\text{sortpool}$ and $\text{maxpool}$ are *normalization-equivariant* functions, Lemma 2 states that the resulting function $f_\theta^{\text{NE}}$ is also *normalization-equivariant*. Moreover, since they are continuous and the assembling operators preserve continuity, $f_\theta^{\text{NE}}$ is continuous. Then, for a given input $x \in \mathbb{R}^n$, we have $(\text{sortpool} \circ a_\Theta)(x) = a_{\pi(\Theta)}(x) = \pi(\Theta)x$, where $\pi$ an operator acting on matrix $\Theta$ by permuting its rows (note that the permutation $\pi$ is both dependent on $x$ and $\Theta$). Therefore, applying a pattern "conv affine + sortpool" simply amounts locally to a linear transformation. Moreover, since applying a max pooling layer amounts to removing some rows from matrix $\Theta$, the local linear behavior is preserved. Thus, as the nonlinearities of $f_\theta^{\text{NE}}$ are exclusively brought by sort pooling patterns (and possibly max pooling layers), $f_\theta^{\text{NE}}$ is actually locally linear. In other words, $f_\theta^{\text{NE}}$ is piecewise-linear. Moreover, as there is a finite number (although high) of possible permutations (and possibly eliminations) of the rows of all matrices $\Theta$, $f_\theta^{\text{NE}}$ has finitely many pieces. Finally, on each piece represented by the vector $y_r$, $f_\theta^{\text{NE}}(y) = A_\theta^{y_r} y$. It remains to prove that $A_\theta^{y_r} \mathbf{1}_n = \mathbf{1}_m$. But this property is easily obtained by noticing that, subject to dimensional compatibility on matrices $\Theta$:

- $\Theta \mathbf{1}_n = \mathbf{1}_m \Rightarrow \pi(\Theta) \mathbf{1}_n = \mathbf{1}_m$ ("conv affine + sortpool"),

- $\Theta \mathbf{1}_n = \mathbf{1}_m \Rightarrow \rho(\Theta) \mathbf{1}_n = \mathbf{1}_l$ ("conv affine + maxpool") where $\rho$ removes some rows,

- $\Theta_1 \mathbf{1}_n = \mathbf{1}_m$ and $\Theta_2 \mathbf{1}_m = \mathbf{1}_l \Rightarrow \Theta_2 \Theta_1 \mathbf{1}_n = \mathbf{1}_l$ (composition),

- $\Theta_1 \mathbf{1}_{n_1} = \mathbf{1}_{m_1}$ and $\Theta_2 \mathbf{1}_{n_2} = \mathbf{1}_{m_2} \Rightarrow \begin{pmatrix} \Theta_1 \\ \Theta_2 \end{pmatrix} \mathbf{1}_{n_1 + n_2} = \mathbf{1}_{m_1 + m_2}$ (skip connection),

- $\Theta_1 \mathbf{1}_n = \mathbf{1}_m$ and $\Theta_2 \mathbf{1}_n = \mathbf{1}_m \Rightarrow (1 - t)\Theta_1 \mathbf{1}_n + t\Theta_2 \mathbf{1}_n = \mathbf{1}_m$ (affine residual connection).

Thus, the affine combinations are preserved all along the layers of $f_\theta^{\text{NE}}$. In the end,

$$f_\theta^{\text{NE}}(y) = A_\theta^{y_r} y, \quad \text{with } A_\theta^{y_r} \in \mathbb{R}^{m \times n} \text{ such that } A_\theta^{y_r} \mathbf{1}_n = \mathbf{1}_m. \qquad \square$$

## C.2 Examples of normalization-equivariant conventional denoisers

**Noise-reduction filters:** All linear smoothing filters can be put under the form $f_\Theta : x \in \mathbb{R}^n \mapsto \Theta x$ with $\Theta \in \mathbb{R}^{n \times n}$ (the rows of $\Theta$ encode the convolution kernel). Obviously, $f_\Theta$ is always *scale-equivariant*, whatever the filter $\Theta$. As for the *shift-equivariance*, a simple calculation shows that:

$$x \mapsto \Theta x \text{ is } \textit{shift-equivariant} \; \Leftrightarrow \; \forall x \in \mathbb{R}^n, \forall \mu \in \mathbb{R}, \Theta(x + \mu \mathbf{1}_n) = \Theta x + \mu \mathbf{1}_m \; \Leftrightarrow \; \Theta \mathbf{1}_n = \mathbf{1}_m \,.$$

Since the sum of the coefficients of a Gaussian kernel and an averaging kernel is one, we have $\Theta \mathbf{1}_n = \mathbf{1}_m$, hence these linear filters are *normalization-equivariant*. The median filter is also *normalization-equivariant* because $\mathrm{median}(\lambda x + \mu) = \lambda \, \mathrm{median}(x) + \mu$ for $\lambda \in \mathbb{R}_*^+$ and $\mu \in \mathbb{R}$.

**Patch-based denoising:**

$-$ NLM [5]: Assuming that the smoothing parameter $h$ is proportional to $\sigma$, *i.e.* $h = \alpha\sigma$, we have $e^{-\frac{\|p(\lambda y_i + \mu) - p(\lambda y_j + \mu)\|_2^2}{(\alpha\lambda\sigma)^2}} = e^{-\frac{\lambda^2 \|p(y_i) - p(y_j)\|_2^2}{\lambda^2 (\alpha\sigma)^2}} = e^{-\frac{\|p(y_i) - p(y_j)\|_2^2}{h^2}}$, hence the aggregation weights are *normalization-invariant*. Then,

$$f_{\mathrm{NLM}}(\lambda y + \mu, \lambda\sigma)_i = \frac{1}{W_i} \sum_{y_j \in \Omega(y_i)} e^{-\frac{\|p(y_i) - p(y_j)\|_2^2}{h^2}} (\lambda y_j + \mu) \text{ with } W_i = \sum_{y_j \in \Omega(y_i)} e^{-\frac{\|p(y_i) - p(y_j)\|_2^2}{h^2}},$$

$$= \frac{1}{W_i} \sum_{y_j \in \Omega(y_i)} e^{-\frac{\|p(y_i) - p(y_j)\|_2^2}{h^2}} \lambda y_j + \frac{1}{W_i} \sum_{y_j \in \Omega(y_i)} e^{-\frac{\|p(y_i) - p(y_j)\|_2^2}{h^2}} \mu \,,$$

$$= \lambda \left( \frac{1}{W_i} \sum_{y_j \in \Omega(y_i)} e^{-\frac{\|p(y_i) - p(y_j)\|_2^2}{h^2}} y_j \right) + \mu \,,$$

$$= \lambda f_{\mathrm{NLM}}(y, \sigma)_i + \mu \,.$$

Finally, $f_{\mathrm{NLM}}$ is a *normalization-equivariant* function.

$-$ NL-Ridge [18]: The block-matching procedure at the heart of NL-Ridge is *normalization-invariant* as it is based on comparisons of the $\ell_2$ norm of the difference of image patches. For each noisy patch group, *a.k.a.* similarity matrix, $Y \in \mathbb{R}^{n \times k}$ composed of $k$ vectorized similar patches of size $n$, the optimal weights $\Theta^* \in \mathbb{R}^{k \times k}$, in the $\ell_2$ risk sense, are computed such that $Y\Theta^*$ is as close as possible to the (unknown) clean patch group $X \in \mathbb{R}^{n \times k}$. The two successive minimization problems approximating $\Theta$ under affine constraints $\mathcal{C} = \{\Theta \in \mathbb{R}^{k \times k}, \Theta^\top \mathbf{1}_k = \mathbf{1}_k\}$ can be put under the form:

$$\Theta^* = \arg\min_{\Theta \in \mathcal{C}} \; \mathrm{tr}\left( \frac{1}{2}\Theta^\top Q\Theta + C\Theta \right) = I_k - n\sigma^2 \left[ Q^{-1} - \frac{Q^{-1}\mathbf{1}_k(Q^{-1}\mathbf{1}_k^\top)}{\mathbf{1}_k^\top Q^{-1}\mathbf{1}_k} \right] \,.$$

with $Q = Y^\top Y$ or $Q = \hat{X}^\top \hat{X} + n\sigma^2 I_k$ for the first and second step, respectively ($\hat{X}$ is the patch group estimate obtained after the first step), $C = n\sigma^2 I_k - Q$ and where $\mathrm{tr}$ denotes the trace operator. Depending on the step, we have:

$$2\,\mathrm{tr}\left( \frac{1}{2}\Theta^\top Q\Theta + C\Theta \right) = \begin{cases} \|Y\Theta - Y\|_F^2 + 2n\sigma^2 \,\mathrm{tr}(\Theta) + \mathrm{const} \\ \text{or} \\ \|\hat{X}\Theta - \hat{X}\|_F^2 + n\sigma^2 \|\Theta\|_F^2 + \mathrm{const} \end{cases}$$

where $\|.\|_F$ is the Frobenius norm. But, for any $Z \in \mathbb{R}^{n \times k}$ and any function $h : \Theta \in \mathbb{R}^{k \times k} \mapsto \mathbb{R}$,

$$\|(\lambda Z + \mu)\Theta - (\lambda Z + \mu)\|_F^2 + n(\lambda\sigma)^2 h(\Theta) = \lambda^2 \left( \|Z\Theta - Z\|_F^2 + n\sigma^2 h(\Theta) \right) \,,$$

assuming that $\Theta^\top \mathbf{1}_k = \mathbf{1}_k$. Therefore, the aggregation weights $\Theta^*$ are *normalization-invariant* and $(\lambda Y + \mu)\Theta^* = \lambda Y\Theta^* + \mu$. Finally, NL-Ridge with affine constraints encodes a *normalization-equivariant* function.

**TV denoising:** Let $y \in \mathbb{R}^n$, $\lambda \in \mathbb{R}_*^+$ and $\mu \in \mathbb{R}$. Let $x^* = \arg\min_{x \in \mathbb{R}^n} \|x\|_{\mathrm{TV}}$ s.t. $\|y - x\|_2^2 = n\sigma^2$ be the solution of TV [37].

$$f_{\mathrm{TV}}(\lambda y + \mu, \lambda\sigma) = \arg\min_{x \in \mathbb{R}^n} \|x\|_{\mathrm{TV}} \quad \text{s.t.} \quad \|\lambda y + \mu - x\|_2^2 = n(\lambda\sigma)^2 \,,$$

$$= \underset{x \in \mathbb{R}^n}{\arg\min} \; \lambda \left\| \frac{x - \mu}{\lambda} \right\|_{\mathrm{TV}} \quad \text{s.t.} \quad \lambda^2 \left\| y - \frac{x - \mu}{\lambda} \right\|_2^2 = \lambda^2 n \sigma^2 \,,$$

$$= \underset{x \in \mathbb{R}^n}{\arg\min} \; \left\| \frac{x - \mu}{\lambda} \right\|_{\mathrm{TV}} \quad \text{s.t.} \quad \left\| y - \frac{x - \mu}{\lambda} \right\|_2^2 = n \sigma^2 \,,$$

$$= \lambda x^* + \mu \,,$$

$$= \lambda f_{\mathrm{TV}}(y, \sigma) + \mu \,.$$

Finally, $f_{\mathrm{TV}}$ is a *normalization-equivariant* function.

## D   Additional results

Table 5: Real-world image denoising on Darmstadt Noise Dataset (DND) with raw data and variance-stabilizing transformation (VST). All "non-blind" methods were trained solely on synthetic white Gaussian noise.

| | Quality metric | PSNR ↑ | SSIM ↑ | | Quality metric | PSNR ↑ | SSIM ↑ |
|---|---|---|---|---|---|---|---|
| | *ordinary* | 47.58 | 0.9762 | | *ordinary* | 47.37 | 0.9754 |
| DRUNet | *scale-equiv* | 47.59 | 0.9763 | FDnCNN | *scale-equiv* | 47.31 | 0.9754 |
| | **norm-equiv** | 47.57 | 0.9762 | | **norm-equiv** | 47.35 | 0.9753 |

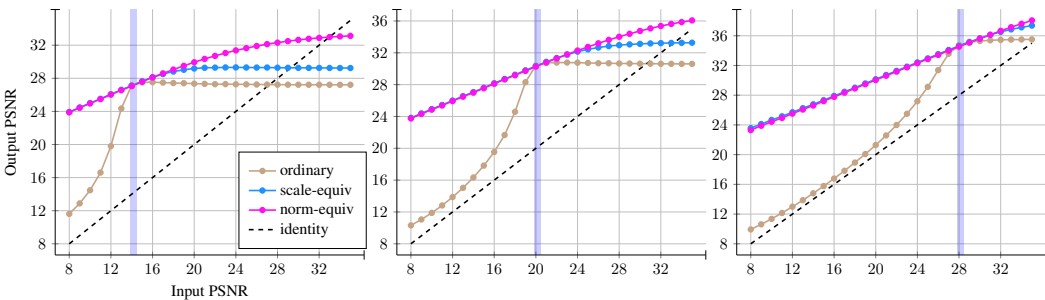

Figure 6: Comparison of the performance of our *normalization-equivariant* FDnCNN [47] with its *scale-equivariant* and *ordinary* counterparts for Gaussian denoising on the Set12 dataset. The vertical blue line indicates the unique noise level on which the networks were trained exclusively (from left to right: $\sigma = 50$, $\sigma = 25$ and $\sigma = 10$). In all cases, *normalization-equivariant* networks generalize much more robustly beyond the training noise level.

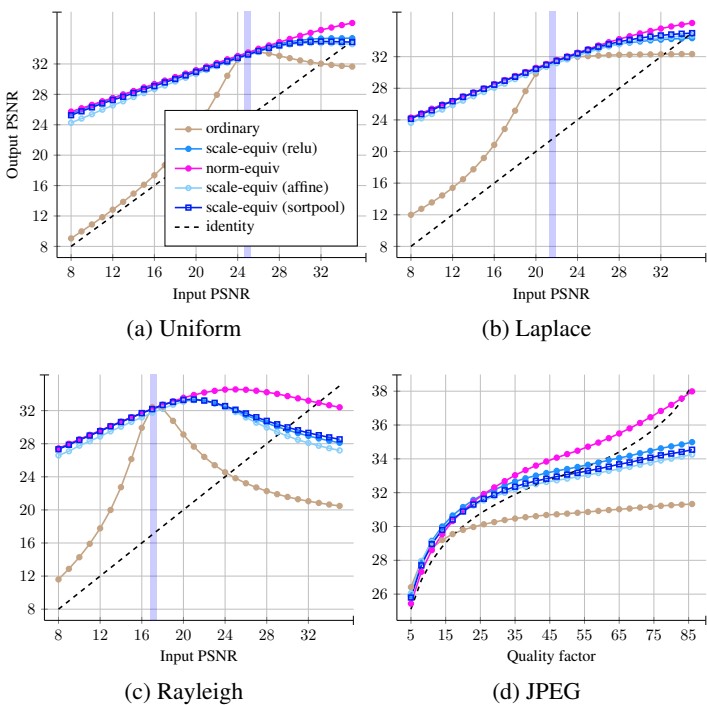

(a) Uniform

(b) Laplace

(c) Rayleigh

(d) JPEG

Figure 7: Comparison of the performance of our *normalization-equivariant* alternative with its *scale-equivariant* (under three different forms) and *ordinary* counterparts for different types of additive noises on the Set12 dataset with "blind" FDnCNN architecture. The vertical blue line indicates the unique noise level on which the "blind" networks were trained exclusively. Note that JPEG noise (d) (i.e. JPEG artifacts) is treated with the networks of (a), assimilating JPEG noise with uniform noise (similar outcomes can be achieved when using the networks specialized for Laplace noise (b)).

Table 6: Test error of different variants of the same VGG8b architecture for image classification. Note that norm-equivariance is with respect to the classification vector and becomes norm-invariance after label selection via argmax. See [34] for details about architecture, datasets and training.

|  | Dataset | MNIST | Kuzushiji-MNIST | Fashion-MNIST |
|---|---|---|---|---|
| | *ordinary* | 0.26% | 1.53% | 4.53% |
| VGG8b | *scale-equiv* | 0.37% | 1.52% | 5.01% |
| | ***norm-equiv*** | 0.44% | 2.78% | 6.57% |

Note that for ***norm-equiv*** variants, learning rate is initialized to $3e\text{-}5$ instead of $5e\text{-}4$ and dropout rate is halved.

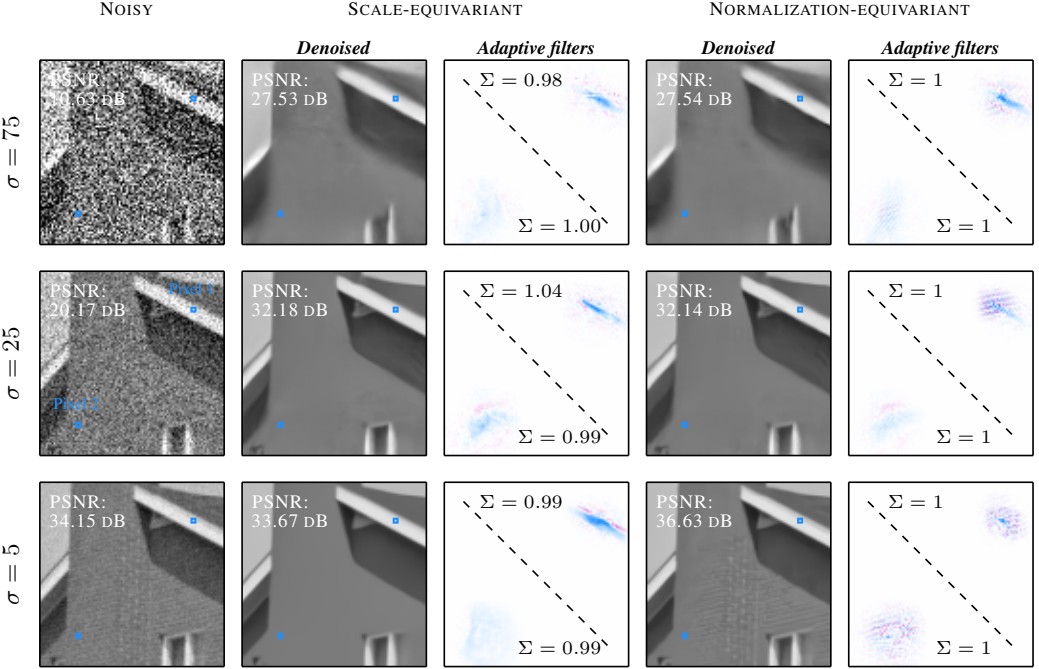

Figure 8: Visual comparisons of the generalization capabilities of a *scale-equivariant* FDnCNN [47] (left) and its *normalization-equivariant* counterpart (right) for Gaussian noise. Both networks were trained for Gaussian noise at noise level $\sigma = 25$ exclusively. The adaptive filters (rows of $A_\theta^{y_r}$ in Prop. 3) are indicated for two particular pixels as well as the sum of their coefficients (note that some weights are negative, indicated in red). The *scale-equivariant* network tends to excessively smooth out the image when evaluated at a lower noise level, whereas the *normalization-equivariant* network is more adaptable and considers the underlying texture to a greater extent.

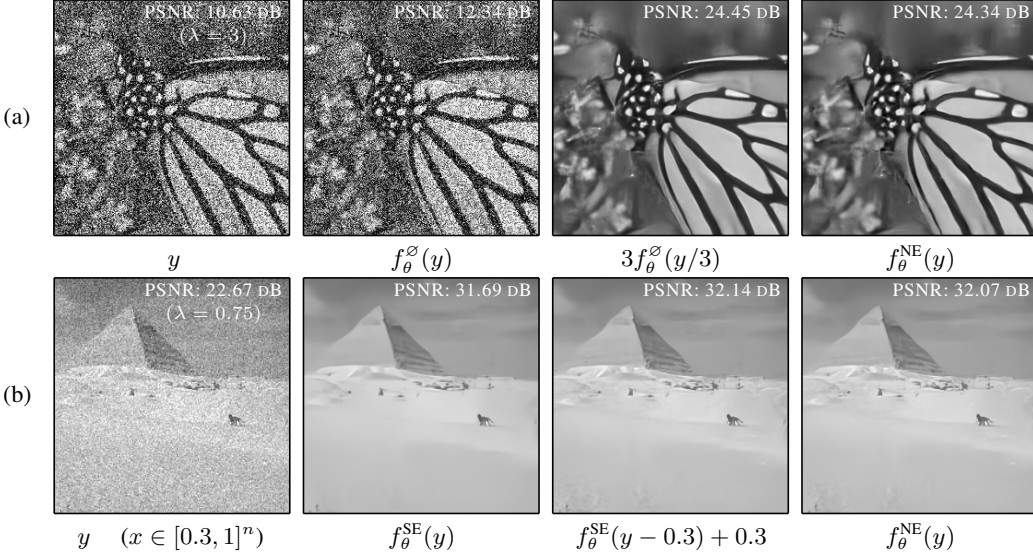

Figure 9: Denoising results for example images of the form $y = x + \lambda\varepsilon$ with $\sigma = 25/255$ and $x \in [0, 1]^n$, by FDnCNN [47] specialized for noise level $\sigma$ only. Here, $f_\theta^\varnothing$, $f_\theta^{\text{SE}}$ and $f_\theta^{\text{NE}}$ denote the *ordinary*, *scale-equivariant* and *normalization-equivariant* variants, respectively.

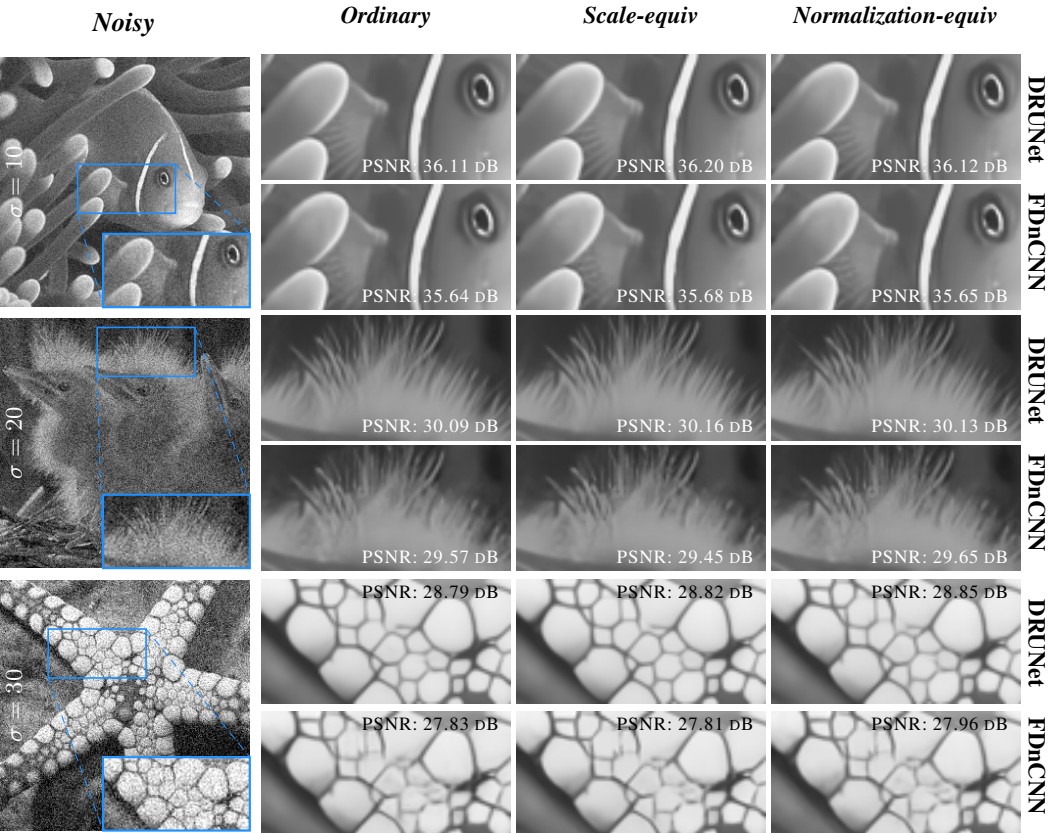

Figure 10: Qualitative comparison of image denoising results with synthetic white Gaussian noise for "non-blind" models. Regardless of the variant of a model, the denoising results are visually similar.

