# OpenReview forum: "Normalization-Equivariant Neural Networks with Application to Image Denoising"
_NeurIPS.cc/2023/Conference — NeurIPS 2023 poster_

### Official Review · Reviewer_RNG3 · 2023-07-05

**Soundness:** 3 good
**Presentation:** 3 good
**Contribution:** 2 fair
**Rating:** 5
**Confidence:** 3

**Summary:**

The paper proposes normalization-equivariant CNNs, in which the convolutional kernels are forced to sum to 1 and the activation sorts pairs of neurons. This is shown to perform similarly to other models on image denoising when trained with the same noise levels, but generalizes much better to unseen noise levels.


**Strengths:**

S1) The paper presents a simple and effective method to improve generalization in image denoising CNNs.

S2) Results are strong, particularly the ones shown in Fig 4.

S3) Section 5.2 presents insights about previously observed phenomena of bias-free/scale-invariant models performing well on denoising, and explains why the normalization-equivariance proposed is potentially superior.


**Weaknesses:**

W2) Table 2 reports results on the non-blind experiments but I couldn't find whether the results in Fig 4 and 5 are blind or non-blind. I assume they are blind and since the models were trained on a single noise level. If they are not and if the noise level is known at inference time, I think the significance of the results is diminished since with a known noise level the ordinary CNNs can be perform quite well as shown in Fig. 5. Please clarify.

W3) Fig 2 implies that the kernel constraint for convolutional kernels is along the channel axis, but it seems that it should be along both channel and spatial dimensions to preserve shift-equivariance. The text does not seem to specify it for the convolutional kernel since Eq 5 refers to a linear layer. Please clarify and include a precise definition of the convolutional kernel between two layers of shape H x W x C.

W4) Some recent methods of denoising are missed [1, 2]. Zamir et [1] seems to report slightly better results than the baselines on denoising on the same datasets and also evaluate on different datasets. This is not a dealbreaker but the submission would be made stronger when evaluating on more datasets since the ones used are quite small.

References:

[1] Zamir et al, "Restormer: Efficient Transformer for High-Resolution Image Restoration", CVPR'22.

[2] Chen et al, "Simple Baselines for Image Restoration", ECCV'22.


**Questions:**

Minor:
Typically in image processing, for an image f: Z^2 -> R, "scale" refers to resizing (f'(x) = f(ax)) and "shift" refers to moving pixels (f'(x) = f(x+a), in the sense that convolution is shift-equivariant). In the submission, it seems that "scale" means multiplying all pixel values by a constant and shift means adding a constant to all pixels. I recommend clarifying the difference to avoid confusion.


**Limitations:**

W1) I couldn't find a discussion of limitations in the paper and I think this is a major weakness. Table 1 in the supplement shows that the norm-equivariant models need to be trained for about twice the number of iterations of the baselines and with MSE instead of L1 loss. For fairness, I think the ordinary and scale-eq baselines should be also evaluated under the same conditions and this possible limitation should be reported in the main text. It seems that the method is also incompatible with other common used layers such as max-pooling and batch normalization, which should also be discussed.

---

> ### Author Rebuttal · Authors · 2023-08-02
>
> W2. They are “blind” indeed since the models were trained on a single noise level (and the noise level is not known at inference time). For the sake of clarity, we will explicitly mention it in the legend of Figures 4 and 5.
>
> W3. Thank you for pointing this out. The transition from a dense layer (5) to a convolutional layer is explained in the paragraph on line 157, perhaps too quickly. The key observation is that a convolution from an input layer of size $H \times W \times C$ to an output layer of size $H' \times W' \times C'$ can always be represented by a dense layer by vectorizing the input and output layers: the elements of the $C'$ convolutional kernels of size $k \times k \times C$ each are then stored separately along the rows of the (sparse) transition matrix $\Theta$ of size $(H' \times W' \times C') \times (H \times W \times C)$. Therefore, the sum of the elements of each kernel of size $k \times k \times C$ must sum up to $1$ for enforcing the normalization-equivariance property. We will add a clarification of that point in the manuscript.
>
> W4. The methods [1, 2] that you mention are both SOTA transformers (in the sense that they are based on the attention mechanism) in the same way as SwinIR [3] is a SOTA transformer, which we have cited in the manuscript. The performance differences between these three networks [1,2,3] and SOTA fully convolutional network DRUNet [5] are marginal, from our point of view, but, for the sake of completeness, we will add these two references to the manuscript.
>
> However, we want to emphasize that the point of our paper is not to introduce a new architecture outperforming its predecessors on some datasets but only to propose architectural modifications to existing neural networks that enforce normalization-equivariance by design, which can be beneficial for robustness against unseen noise levels, in addition to their better conditioning. For the sake of completeness, we have evaluated the three “non-blind” network variants on a supplementary dataset composed of real-world noisy images [4] on which a variance-stabilizing transformation (VST) was applied beforehand and draw the same conclusion: the performance remains the same at already seen noise levels between two different variants (see Table 1 of attached pdf). Please note that the drop of performance on this dataset compared to the methods you mention [1, 2] is mainly due to the fact that the three network variants were trained solely on synthetic white Gaussian noise.
>
> Q1. Thank you for pointing this out. In line 63 of the manuscript, we have added: “Please note that our definition of "scale" and "shift" may differ from the definition given by some authors in the image processing literature”.
>
> W1. Regarding the limitations of the paper, please refer to the global response to reviewers. Note that our method is compatible with max pooling since the max function is normalization-equivariant (by the way, the proposed sort pooling layer can be implemented using one-dimensional max pooling and min pooling). As for the difference in loss functions for training (MSE / L1), we decided to take the best-performing loss function for each method, as explained in the Supplementary Material. As it happens, normalization-equivariant networks obtain slightly better results with the usual MSE loss (but the L1 loss is also possible). On the other hand, the authors  [5, 6] recommend the L1 loss to achieve better performance for the original networks DRUNet and FDnCNN and we followed strictly their instructions for training. Finally, we trained all models to convergence, as in [5, 6], and it turns out that normalization-equivariant networks need longer training to reach convergence. This specificity will be reported in the Limitation section we plan to write.
>
> [3]  Liang et al., “SwinIR: Image Restoration Using Swin Transformer,” ICCVW'21.
>
> [4] Plötz and Roth, “Benchmarking Denoising Algorithms with Real Photographs,” CVPR'17.
>
> [5] Zhang et al., “Plug-and-play image restoration with deep denoiser prior,” IEEE Trans PAMI 2021.
>
> [6] Zhang et al., “Beyond a Gaussian denoiser: residual learning of deep CNN for image denoising,” IEEE Trans IP 2017.

---

> > ### Comment · Reviewer_RNG3 · 2023-08-14
> >
> > I appreciate the response, which resolve my points W2, W3, W4. I still have reservations about the use  of MSE/L1 and longer training for the proposed method. I believe comparison with the baselines should be done under the same conditions, to ensure that the better results come from the paper's contributions and not choice of loss/training schedule. I recommend also training the baseline using the same loss and number of iterations to remove this doubt.

---

> > > ### Author Response · Authors · 2023-08-15
> > >
> > > Thank you for your feedback.
> > >
> > > Regarding your last point, if we train our "non-blind" normalization-equivariant networks with the setting recommended for the other variants ($\ell_1$ loss and fewer iterations), we observe a small PSNR drop of 0.15 dB for both networks (FDnCNN and DRUNet) for $\sigma=25$ on Set12 dataset for example.
> > > Even if adapting the training setting to each architecture variant is acceptable from our point of view, we will work on this in the coming weeks to see if this difference can be somehow reduced.
> > >
> > > Please note, however, that the difference of training setting does not explain the better generalization capabilities to unseen noise levels of the "blind" normalization-equivariant networks. The curves of Figure 1 of the attached pdf were realized under exact same training setting for all variants for example ($\ell_1$ loss and fewer iterations) because we did not look for performance; and the curves in the paper for Gaussian noise stay the same for both training settings (the difference is almost imperceptible to the eye).

---

### Official Review · Reviewer_6aGn · 2023-07-05

**Soundness:** 4 excellent
**Presentation:** 3 good
**Contribution:** 2 fair
**Rating:** 6
**Confidence:** 4

**Summary:**

The authors of this paper propose an architectural change in neural networks, in a way that enforces the normalization equivariance property. The authors demonstrate how networks that satisfy this property can be used in the setting of image denoising.

**Strengths:**

- The paper is clearly written overall. The intended goal of the architectural changes proposed is clear, and how these changes achieve this goal is also made explicit in the paper. The proposed architecture is also novel, as far as I am aware.

- There is a clear benefit derived from the architectural changes proposed by the authors. The intended effect of normalization equivariance is to make the output of the network disregard potential alterations in the scale of the noise added to the model. This is made clear via Figure 4, which precisely shows that the alterations to the architecture achieve the intended effect. The authors also adequately explain in the paper how this equivariance with respect to this normalization makes their models robust to the amount of noise added to the input, namely in that adding different amounts of noise only corresponds to renormalization.

- The idea that the authors propose regarding sort pool between channels is an interesting and novel way of pooling information from different channels in CNNs. This seems to be a novel inclusion in the architecture of a neural network, and I am interested in its potential applications beyond normalization equivariance.


**Weaknesses:**

My main issues with this paper lie with the somewhat limited setting of the experiments performed to validate the abilities of the proposed architecture.

- While interesting, the target problem for this paper is somewhat limited. The only application considered in this paper is that of image denoising, which is arguably a very limited setting. I believe that demonstrating the effects of enforcing normalization equivariance in other problems (such as image classification or other types of reconstruction e.g. superresolution) would improve the paper.

- Even in the context of denoising, the authors of this paper only consider the setting of Gaussian noise being applied to an image. While generally accepted as a valid setting for denoising, It would be interesting to see if the normalization equivariance was able to help in the case where different types of noise are applied (e.g. salt-pepper noise, random masking of pixels, and so on).

- Regarding baselines, the authors compare against techniques which use regular feedforward neural networks. While the comparisons support the points made by the authors, I believe that it would be interesting to compare against techniques that employ generative networks to solve the denoising problems e.g. via DPS [A].

Reference:

[A] Chung et al., “Diffusion Posterior Sampling for General Noisy Inverse Problems”, ICLR 2023.


**Questions:**

I would be interested to know if the authors have considered applying their architectural change not when training the models from scratch, but by altering select layers of a pretrained model (for example, by including the sorting layer as part of a pretrained architecture). If something like this is possible, then it would alleviate some of the issues I mentioned above, allowing the method to be more readily applied to a larger set of problems.

**Limitations:**

I believe the authors should better clarify in the paper how currently their method is applicable mainly to image denoising as it stands (even though in principle normalization equivariance may be helpful in other tasks as well).

Regarding negative societal impact, I do not see any immediate issues arising from this work.

---

> ### Author Rebuttal · Authors · 2023-08-02
>
> Regarding your reservations about paper limitations, please refer to the global response to reviewers.
>
> - “the authors compare against techniques which use regular feedforward neural networks. While the comparisons support the points made by the authors, I believe that it would be interesting to compare against techniques that employ generative networks to solve the denoising problems e.g. via DPS [A].”
>
> To the best of our knowledge, the methods that employ generative networks such as [A] in order to solve inverse problems are superior to regular feedforward neural networks only in the case of severe degradation models, for which the usual principle of empirical risk minimization is helpless to recover realistic attributes (e.g., full face inpainting, extreme superresolution, or very high noise...). Posterior sampling is then an interesting alternative. As far as we are aware, the SOTA methods for non-extreme noise removal are all regular feedforward neural networks.
>
> - “I would be interested to know if the authors have considered applying their architectural change not when training the models from scratch, but by altering select layers of a pretrained model (for example, by including the sorting layer as part of a pretrained architecture). If something like this is possible, then it would alleviate some of the issues I mentioned above, allowing the method to be more readily applied to a larger set of problems.”
>
> This is indeed an interesting line of research. Preliminary results suggest that no gain in terms of training time can be achieved by retaining all the pre-trained weights. However, further investigations are needed to find out if it is possible to keep only carefully selected weights and it could be an interesting line of research for future work.

---

> > ### Comment · Reviewer_6aGn · 2023-08-11
> > **Response to author rebuttal.**
> >
> > I thank the authors for their response to my concerns.
> >
> > From the general response, I see that my concern regarding the use of only Gaussian noise as a setting has been alleviated, with experiments showing robustness to other types of noise as well.
> >
> > I also appreciate the effort the authors made to extend their proposed method to settings beyond image denoising. While simple, the image classification experiments provided show some preliminary results on how the method can be used on tasks other than denoising. This somewhat alleviates the limitations of the paper. As such, I have raised my score slightly.

---

> > > ### Author Response · Authors · 2023-08-15
> > >
> > > Thank you for your positive feedback

---

### Official Review · Reviewer_dbvT · 2023-07-06

**Soundness:** 4 excellent
**Presentation:** 4 excellent
**Contribution:** 3 good
**Rating:** 8
**Confidence:** 5

**Summary:**

In this paper, the authors propose an architectural design which guarantees shift and scale equivariance. The proposed method is called normalization equivariance in which scale equivariance is inherited from prior work by removing layers biases and shift equivariance is achieved through affine-constrained convolutions along with channel-wise sort pool (instead of ReLUs). They adopt two popular architectures to become normalization equivariance, and show that their variant outperforms both unconstrained, and scale-equivariance models in generalizing outside of the training range.

**Strengths:**

The paper is written very clearly and has excellent flow. It is technically strong and propositions and claims are backed up both theoretically and empirically. The idea of enforcing shift equivariance by design is original and the authors nicely elucidate its necessity and the positive impact it has. In my opinion, this work offers a significant contribution in the area of useful inductive biases and also understanding neural networks trained for denoising.

**Weaknesses:**

Generally the paper is written very well and all sections are carefully and throughly explained. Here are a few minor issues:

I believe rotation needs to generally be an invariance (as opposed to equivariance) with respect to labels in image classification (line 52)

Proof for proposition 2 can be included in the appendix.

The derivation of eq 6 (from line 178 onward) is not clear. It would be useful if the authors re-write that paragraph.

Number of possible states is still $2^S$ if ReLU activations are used instead of sorting (line 222)

**Questions:**

It would be useful to see comparison of computational cost for training. Although sort pooling is restricted to n=2, I assume it is still most costly than using ReLUs. A numeral comparison can show the cost for gaining normalization-equivariance.

It wasn't clear from the text how the affine convolution constraint was enforced during training. It would be useful to explain that in the main text since it is one of the two main design contributions.

**Limitations:**

No major limitations or potential negative societal impacts.

---

> ### Author Rebuttal · Authors · 2023-08-02
>
> Minor issues:
>
> - “I believe rotation needs to generally be an invariance (as opposed to equivariance) with respect to labels in image classification (line 52)”.
>
> Thank you for pointing this out. We agree with you, this was an inaccuracy and we removed it from the manuscript.
>
> - “Proof for proposition 2 can be included in the appendix.”
>
> The proof can be found from line 112 to line 122 in the supplementary material, even though it was not mentioned in the manuscript. For the sake of clarity, we have added “(see proof in Supplementary Material)” to line 78 of the manuscript.
>
> - “The derivation of eq 6 (from line 178 onward) is not clear. It would be useful if the authors re-write that paragraph.”
>
> We assume you are referring to the transition from the arbitrarily setting $\varphi(-u) = \varphi(u) = u$ to the deduction that $\varphi$ is the sorting function of $\mathbb{R}^2$. For greater clarity, we have added the following explanation to line 185: “Indeed, the sorting function (6) is a normalization-equivariant function by concatenation (see Prop. 2) of two normalization-equivariant functions from $\mathbb{R}^2$ to $\mathbb{R}$, namely the functions min and max. Moreover, it does verify $\varphi(-u) = \varphi(u) = u$ and this is the only one according to Prop. 1. Note that another choice of arbitrary setting would have entirely identified a different normalization-equivariant function of $\mathbb{R}^2$, possibly leading to an unintended linear function or to a more complicated nonlinear function.” To be perfectly exact, from the proof of the existence (line 97 of the Supplementary Material), one can show that all normalization-equivariant functions of $\mathbb{R}^2$ to $\mathbb{R}^2$ are under the form: $\varphi: (x_1, x_2) \mapsto \frac{1}{\sqrt{2}} | x_1 - x_2 | y_{\operatorname{sign}(x_1 - x_2)  u} + \frac{1}{2}  (x_1+x_2, x_1+x_2)$  where $u=(-1/ \sqrt{2}, 1/ \sqrt{2})$ and $y_{-u} \in \mathbb{R}^2$ and $y_u \in \mathbb{R}^2$ are to be chosen. Clearly, the sorting function is the simplest nonlinear function of this form, obtained for $y_{-u}=y_u=u$.
>
> - “Number of possible states is still $2^{S}$ if ReLU activations are used instead of sorting (line 222)”
>
> Unless we are mistaken, using ReLU activation functions instead of sort pooling nonlinearities generally allows for a greater number of pieces in the resulting piecewise-linear functions. It can be seen by considering the following rudimentary feedforward neural network $f_\theta : (x, y) \in \mathbb{R}^2 \mapsto  \Theta_2 \varphi\left(\Theta_1 (x \quad y)^\top \right)$ with $\varphi$ the sorting function (6), $\Theta_1\in\mathbb{R}^{2\times2}$ and $\Theta_2\in\mathbb{R}^{1\times2}$. In this case, $S=1$. An upper bound for the total number of pieces is then $2^{S}=2$ which is attained for $\Theta_1=I_2$ and $\Theta_2=(-1\quad1)$ but not for $\Theta_2=(0\quad0)$, in which case $f_\theta$ is the zero function (only one piece). On the other hand, if $\varphi$ is the ReLU function, then an upper bound is $2^{2S}=4$ which is attained for the same parameters $\Theta_1=I_2$ and $\Theta_2=(-1\quad1)$. (a software can help for visualization)
>
> Questions:
>
> - “It would be useful to see comparison of computational cost for training. Although sort pooling is restricted to $n=2$, I assume it is still most costly than using ReLUs. A numeral comparison can show the cost for gaining normalization-equivariance.”
>
> You are right, it is more costly to gain normalization-equivariance. We have compared the computational costs of different variants for training and inference (see Table 3 of attached pdf). Interestingly, the computational cost for training is much more sensitive to the “affine mode” (involving affine-constrained convolutions with reflect padding and affine residual connection) than to sort pooling nonlinearities, while it is the opposite for inference. All in all, for gaining normalization-equivariance, the learning and inference time is almost doubled for the DRUnet architecture on a Quadro RTX 6000 GPU. Note however that we do not claim to have the most optimized implementation and there is probably room for improvement.
>
> - “It wasn't clear from the text how the affine convolution constraint was enforced during training. It would be useful to explain that in the main text since it is one of the two main design contributions.”
>
> We understand your point. Due to space limitation, we preferred to defer the description of the practical implementation in the supplementary material. However, if the article is accepted and if any space remains, we plan to raise the issue in the main text.

---

> > ### Comment · Reviewer_dbvT · 2023-08-17
> >
> > Thank you for responding to my questions and comments. I recommend you include Fig 1 and Table 3 in the paper (or in the appendix).

---

> > > ### Author Response · Authors · 2023-08-17
> > >
> > > Thank you for your positive feedback. We will follow your recommendations and include them in the paper.

---

### Official Review · Reviewer_atAH · 2023-07-07

**Soundness:** 4 excellent
**Presentation:** 4 excellent
**Contribution:** 3 good
**Rating:** 5
**Confidence:** 4

**Summary:**

This work proposes a normalization equivariant deep convolutional neural network for image denoising. The authors provide a mathematical construction of a normalization equivariant neural network and propose sorting nonlinearity. The proposed nonlinearity shows comparable performance with the baseline and shows robustness against different noise levels.

**Strengths:**

1. The work provides a sound mathematical formulation for the Normalization equivariant Neural Network.
2. The proposed model is robust across different noise level
3. The paper is well written, nicely explained

**Weaknesses:**

1. The effect of newly introduced sorting nonlinearity will be better understood through an ablation study, i.e., reporting the performance of a model with only affine convolution and Relu.
2. The effect of shift-equivariance is not well described and demonstrated. Even though it is addressed in the last section, its’ benefit over the scale-equivariant model is not well demonstrated and motivated. Given the novelty of the work is providing shift equivariance (in addition to scale-equivariance), it deserves better motivation and justification.
3. The proposed non-linearity performs point-wise swapping of the output of different filters at some points. So, it distorts the learned features. It has a significant possibility of affecting the performance of other computer vision tasks (e.g., segmentation), limiting its use only to denoising.

**Questions:**

1. The work [1] shows a better generalization of scale equivariant neural networks on different noise scales compared to what is shown here. What might be different here that lead to this difference?
2. Line 282, “but this is much less true for λ < 1 (stretching) for the reason that it may exceed the bounds of the interval [0, 1]. This explains why scale-equivariant functions do not generalize well to noise levels lower than their training one”: how λ < 1 related to not generalizing to lower noise levels?

1.ROBUST AND INTERPRETABLE BLIND IMAGE
DENOISING VIA BIAS-FREE CONVOLUTIONAL
NEURAL NETWORKS

---

> ### Author Rebuttal · Authors · 2023-08-02
>
> W1. First of all, we would like to stress that our aim is not to introduce new layers that might perform better than traditional ones, but only to observe that it is possible to use the proposed substitutes as a viable alternative that allows normalization-equivariance, without degrading performance on already observed noise levels, which is demonstrated in Table 2 of the paper. In fact, our point is simply that affine convolutions and ordinary convolutions are interchangeable, as are sort pooling nonlinearities with ReLU. However, the benefits in terms on performance of the proposed architectural modifications are revealed at unseen noise levels, as demonstrated in Figure 4 of the paper. If, however, your question was: “Are affine convolutions alone or channel-wise sort-pooling nonlinearities alone sufficient to generalize to unseen noise levels?”, the answer is negative (see Figure 1 of attached pdf). The reason is because only the strict association of affine convolutions with sort-pooling nonlinearities allows normalization-equivariance and it is this property that allows increased robustness to unseen noise levels as theoretically proven (see line 268).
>
> W2-W3. Regarding the limitations of the paper, please refer to the global response to reviewers. Note that premilinary results suggest that normalization-equivariant networks (implemented using channel-wise sort pooling layers) are applicable to image classification (see Table 2 of attached pdf).
>
> Q1. “What might be different here that lead to this difference?”
>
> In our experiments, we trained all “blind” models exclusively on a single noise level (either $\sigma=10, 25$ or $50$), contrary to [1] that considers whole ranges of noise levels, which may explain this difference. Moreover, in their experiments they mainly train the “blind” models on ranges of the form $[0, \sigma_{\operatorname{max}}]$ ($\textit{e.g.}$, the noise ranges are $[0, 10]$, $[0, 30]$, $[0, 55]$ and $[0, 100]$ in the code they provided). Therefore, the drop of performance of scale-equivariant networks at lower noise levels cannot be observed ($\textit{e.g.}$, see Figure 11 of [1]). Finally, the drop of performance for scale-equivariant networks is even more pronounced with the DRUNet architecture for some reason, but this architecture appeared after the publication of [1].
>
> Q2. “How $\lambda < 1$ related to not generalizing to lower noise levels?”
>
> In the case of a scale-equivariant network $f_\theta$, we have, adopting the same notations as in subsection 5.2:   $f_\theta(x + \lambda \varepsilon) = \lambda f_\theta(x/\lambda + \varepsilon)$. Hence, denoising the image patch $x + \lambda \varepsilon$ amounts to denoising $x/\lambda + \varepsilon$. Interestingly, $x/\lambda + \varepsilon$ is a patch corrupted by the exact type of noise on which $f_\theta$ has been trained for. Therefore, as long as $x/\lambda$ is in $\mathcal{D} \subset [0, 1]^n$, efficient denoising can be guaranteed. In general, $x \in \mathcal{D} \Rightarrow x/\lambda \in \mathcal{D}$ for $\lambda > 1$ (think of a flat patch $x$ for example) but this is much less true for $\lambda < 1$ for the reason that it may exceed the bounds of the training interval $[0, 1]$. Indeed, if $x=1_n$ for example (flat white patch) and $\lambda=0.5$ (half the amount of noise for which $f_\theta$ has been trained), then $x/\lambda = 2_n$ which is not in  $[0, 1]^n$, therefore there is no guarantee as to the result of the denoising of patch $(x/\lambda + \varepsilon)$ by $f_\theta$. But if $f_\theta$ is in addition shift-equivariant (i.e., normalization-equivariant), then $f_\theta(x/\lambda + \varepsilon) = f_\theta((x/\lambda - 1_n) + \varepsilon) + 1_n$, which amounts to denoising the clean patch $(x/\lambda - 1_n) = 2_n - 1_n = 1_n$ corrupted by the exact type of noise on which $f_\theta$ has been trained for. This time $(x/\lambda - 1_n)$ is in $\mathcal{D} \subset [0, 1]^n$ and we have then a guarantee as to the result of the denoising. This behavior on flat patches was confirmed experimentally.

---

> > ### Comment · Reviewer_atAH · 2023-08-16
> > **Response to the Rebuttal**
> >
> > I thank the authors for the response.

---

### Author Rebuttal · Authors · 2023-08-02

First of all, we want to thank the reviewers for dedicating their valuable time to the evaluation of our paper and for providing us with constructive feedback, which will improve the quality and clarity of the manuscript. We have revised the manuscript, taking into account all the comments received.

Criticism from some reviewers focused on the possibility of applying the proposed normalization-equivariant neural networks to computer vision tasks other than denoising, such as classification, segmentation or super-resolution. Preliminary results on image classification indicate that it is indeed possible (see Table 2 of attached pdf). However, we focused in this paper on the applications of normalization-equivariant networks on image denoising because, beyond their better conditioning that avoids a major source of confusion and misinterpretation linked to normalization, such networks proved to be exceptionally robust against unseen noise levels, which is a remarkable benefit specific to image denoising. Clearly, discovering similar advantages of normalization-equivariance in other computer vision tasks, possibly related to outlier robustness, is an interesting avenue of research for future work.

In the context of image denoising, several reviewers point out that the paper only considers the setting of Gaussian noise for the experimental results, which is the most widely studied image noise model. In fact, it turns out that our theoretical demonstration (line 268 of the paper) of the increased robustness across noise levels is still valid for a wide range of noise types. Indeed, our demonstration remains valid for any additive noise $\varepsilon$ that possesses the scaling property: $\forall \lambda > 0, \lambda \varepsilon$ belongs to the same family of probability distributions as $\varepsilon$ (e.g., Gaussian noise: $\varepsilon \sim \mathcal{N}(0, \sigma^2) \Rightarrow \lambda \varepsilon \sim \mathcal{N}(0, (\lambda\sigma)^2)$, uniform noise $\varepsilon \sim \mathcal{U}(-a, a) \Rightarrow \lambda \varepsilon \sim \mathcal{U}(-\lambda a, \lambda a)$, Laplace noise $\varepsilon \sim \mathcal{L}(0, b) \Rightarrow \lambda \varepsilon \sim \mathcal{N}(0, \lambda b)$ or even Rayleigh noise which is not zero-mean: $\varepsilon \sim \mathcal{R}(\sigma) \Rightarrow \lambda \varepsilon \sim \mathcal{R}(\lambda\sigma)$). By the way, the authors of [1] had already verified the noise generalization capabilities of scale-equivariant networks for uniform noise in addition to Gaussian noise, without fully elucidating why it works. In the attached pdf, we checked experimentally that “blind” normalization-equivariant networks trained on additive uniform, Laplace or Rayleigh noise at a single noise level are much more robust at unseen noise levels than their scale-equivariant and ordinary counterparts. Moreover, we also observed the robustness capabilities of normalization-equivariant networks when testing on a type of noise that is more difficult to characterize, namely JPEG noise, or JPEG artifacts (see Figure 1d of attached pdf).

Finally, we agree with the reviewers that a Limitation section is necessary. If the paper is accepted, we would like to mention that the proposed architectural modifications for enforcing normalization-equivariance do require a longer training for achieving comparable performance with its original counterpart (see Table 3 of attached pdf) , and may be incompatible with some specific network layers such as batch-norm (but compatible with pooling layers, dropout, ...). Moreover, our method has shown its potential mainly to image denoising as it stands (even though in principle normalization-equivariance may be applicable and helpful in other tasks as well).

For further details, please refer to the specific response to each reviewer comment.

[1] S. Mohan et al., “Robust And Interpretable Blind Image Denoising Via Bias-Free Convolutional Neural Networks,” ICLR'19.

---

### Decision · Program_Chairs · 2023-09-21

**Decision:**

Accept (poster)

**Comment:**

The paper presents an effective way to bring normalization-equivariance to NN for image denoising, with a sound mathematical formulation. Before rebuttal, several concerns were raised, e.g., the target problem can be limited, as only image denoising was considered; for denoising, only the setting of Gaussian noise is discussed, etc.  The authors did an excellent job during the discussion phase to address most concerns. All reviewers vote for acceptance.  AC recommends accepting the paper.